# Annual time-series 1-km maps of crop area and types in the conterminous US (CropAT-US): Cropping diversity changes during 1850-2021

Shuchao Ye, Peiyu Cao, Chaoqun Lu

Department of Ecology, Evolution, and Organismal Biology, Iowa State University, Ames, Iowa 5011, USA

*Correspondence:* Chaoqun Lu (clu@iastate.edu)

**Abstract.** Agricultural activities have been recognized as an important driver of land cover/land use change (LCLUC) and have significantly impacted the ecosystem feedback to climate by altering land surface properties. A reliable historical cropland distribution dataset is crucial for understanding and quantifying the legacy effects of agriculture-related LCLUC. While several LCLUC datasets have the potential to depict cropland patterns in the conterminous US, there remains a dearth of a relatively high-resolution dataset with crop type details over a long period. To address this gap, we reconstructed historical cropland density and crop type maps from 1850 to 2021 at a resolution of 1 km×1 km by integrating county-level crop-specific inventory datasets, census data, and gridded LCLUC products. Different from other databases, we tracked the planting area dynamics of all the crops in the US, excluding idle/fallow farm land, and cropland pasture. The results showed that the crop acreages for nine major crops derived from our map products are highly consistent with the county-level inventory data, with the residual less than 0.2 thousand hectares (Kha) in most counties (>75%) during the entire study period. Temporally, the US total crop acreage has increased by 118 million hectares (Mha) from 1850 to 2021, primarily driven by corn (30 Mha) and soybean (35 Mha). Spatially, the hotspots of cropland distribution shifted from Eastern US to the Midwest and the Great Plains, and the dominant crop types (corn and soybean) expanded northwestward. Moreover, we found the US cropping diversity experienced a significant increase from 1850s to 1960s, followed by a dramatic decline in the recent six decades under the intensified agriculture. Generally, this newly developed dataset could facilitate the spatial data development in delineating crop-specific management practices and enable the quantification of cropland change impacts.

## 1 Introduction

Anthropogenic land cover/land use change (LCLUC) has altered nearly 70% of global ice-free land (Arneth et al., 2019), exerting significant effects on ecosystem services by changing biogeochemical and biophysical processes (Foley et al., 2005; Goldewijk et al., 2017; Johnson, 2013; Betts et al., 2007; Lark, 2023). In particular, agricultural activities have been identified as the dominant driver of LCLUC (Cao et al., 2021), with approximately one-third of the land surface altered for agricultural use to meet human demands of food, feed, fiber, and fuel (Zhang et al., 2007). These changes have led to a range of environmental issues, including greenhouse gas emissions (De Noblet-Ducoudré et al., 2012; Yu et al., 2018), agricultural water pollution (Ouyang et al., 2014), and soil degradation (Vanwalleghem et al., 2017). In addition, the intensification of agriculture causes the decline of crop diversity, which can reduce the resilience of crops to various environmental stresses and threaten the crop yield (Burchfield et al., 2019; Gaudin et al., 2015; Renard and Tilman, 2019; Aizen et al., 2019). Therefore, gaining a better understanding of spatiotemporal cropland extent and type changes is critical to quantify the environmental effects of cropland change and promote sustainable agricultural practices (Tilman et al., 2011; Lambin and Meyfroidt, 2011).

As a leading agricultural producer, the conterminous US has experienced a substantial transformation in crop area, distribution, and type over the last two centuries. From 1850s to 1980s, the crop area increased about eightfold from around 20 million hectares to about 160 million hectares, primarily through the conversion of forest, grassland, and other land types (Li et al., 2023; Turner, 1988). Spatially, the development of canals, waterways, and railroads contributed to the cropland expansion to the west (Meinig, 1993). Especially, the Homestead Acts in 1862 played a significant role in stimulating agricultural reclamation. Moreover, in crop commodities, the dominant crop types have shifted. Before the mid-twentieth century, corn and wheat were the dominant crops. However, the cultivated area of soybean has gradually surpassed wheat and became the second widely produced crop type across the US in recent decades (Lubowski et al., 2006). Although these changes have been reported by the government and social scientists (Waisanen and Bliss, 2002), there is still a lack of a long-term cropland dataset to depict the spatial patterns of crop type choice and distribution in the US over a long time period. Despite that long-term crop-specific management information has been available in the US for quite a long period, large uncertainties remain in developing historical management maps and assessing their environmental and economic consequences spatially, because not knowing "what is planted where" is a big hurdle before the remote sensing data is available.

A wide variety of land use datasets have been used to explore the spatiotemporal patterns of agricultural land in the contiguous US. For instance, History database of global environment (HYDE) (Goldewijk et al., 2017) constructed a weighting algorithm involving dynamical social (historical population density and national/sub-national crop statistics, state level crop inventory in US) and stable environmental (soil suitability, temperature, and topography) factors to reconstruct the historical crop distribution at the resolution of 5 arc-minute. Similarly, Zumkehr and Cambell (2013) adopted a land-use model of Romankutty and Foley (Ramankutty and Foley, 1999) and a satellite-derived cropland distribution map to calculate the historical crop area grid by grid under the control of crop inventory records. Although these datasets present the long-term land use change history, their coarse resolutions offer limited spatial details. Growing remote sensing technology and machine learning methods enhance the capability to monitor land surface change with the high resolution LCLUC products (Tian et al., 2014; Shi et al., 2020). For instance, Cropland

Data Layer (CDL), National Land Cover Database (NLCD), and Land Change Monitoring, Assessment, and
Projection (LCMAP) provide the gridded cropland distribution maps at the resolution of 30m by 30m (Homer et al.,
2020; Xian et al., 2022; Lark et al., 2017). However, these high-resolution datasets lack the capability to depict
historical cropland change patterns before the emergence of satellite images. Recently, Cao et al. (2021) harmonized
cropland demands from HYDE and Land-Use Harmonization 2 datasets with the combination of cropland suitability,
kernel density, and other constraints to generate a cropland dataset from 10000 BCE to 2100 CE. Li et al. (2023)
integrated an artificial neural network-based probability of occurrence estimation tool and multiple inventories to
generate the historical cropland maps at the resolution of 1km by 1km. However, the crop type details are still missing
in these datasets, making it challenging to identify the specific crop type change over space and time. On the other
hand, Monfreda et al. (2008) combined a global cropland dataset and multi-level census statistics (national, state, and
county) to generate a map depicting the area and yield of 175 crops circa the year 2000 around the world, and Tang et
al. (2023) further updated it to depict 173 crops circa the year 2020. Their products also provide information that is
only available in the recent two decades, limiting our understanding of historical US crop type development. Overall,
the currently available datasets either have short periods, low spatial resolution, or lack specific crop type information.
This limits our capability in assessing how crop type changes and crop-specific management before 2000 have affected
the climate system and environmental quality at a finer scale. Thus, it is urgent to develop a long-term spatially explicit
cropland dataset with crop type details to comprehend the US agricultural land use history.
In this study, we aim to reconstruct the cropland density and crop type maps in the conterminous US from 1850
to 2021 at 1 km by 1 km resolution. The cropland density maps present the distribution and percentage of crop planting
area in each 1 km by 1 km pixel. The crop type maps display the distribution of nine major crop types (corn, soybean,
winter wheat, spring wheat, durum wheat, cotton, sorghum, barley, and rice) and one category labeled as "others"
(including all remaining crop types but excluding idle/fallow farm land, and cropland pasture). This study consists of
three sections: Section 2 describes the materials and methods used to reconstruct the dataset, Section 3 analyzes the
spatiotemporal changes in dominant crop types and cropping diversity based on the reconstructed dataset, and Section
4 discusses the differences between our dataset and other datasets, the drivers of cropland change, the implications of
US crop diversity change, and the data uncertainty.
**2    Materials and method**
In this study, we combined three inventory datasets and four gridded datasets to reconstruct the historical cropland
density and crop type maps. As illustrated in Figure 1, the entire process involves three stages: reconstructing annual
inventory data for each crop type at the county level (Section 2.2), rebuilding cropland density maps (Section 2.3),
and generating crop type maps (Section 2.4). In particular, we adopted the following assumptions for reconstructing
the cropland maps: (1) the USDA inventory datasets provide the most reliable acreage information for determining
cropland area in each county; (2) Cropland data layer (CDL), History database of the global environment 3.2 (HYDE)
(Goldewijk et al. 2017), and Land change monitoring, assessment, and projection (LCMAP) provide the potential
distribution of cropland, which were used to allocate cropland grids under the control of the rebuilt inventory data (Yu
and Lu, 2018); (3) The rotation percentage between corn and soybean remained constant when the rotation information
was unavailable from 1940 to 2009. Furthermore, based on the generated crop type maps, we explored the historical
US crop diversity pattern through the true diversity index (Jost, 2006).

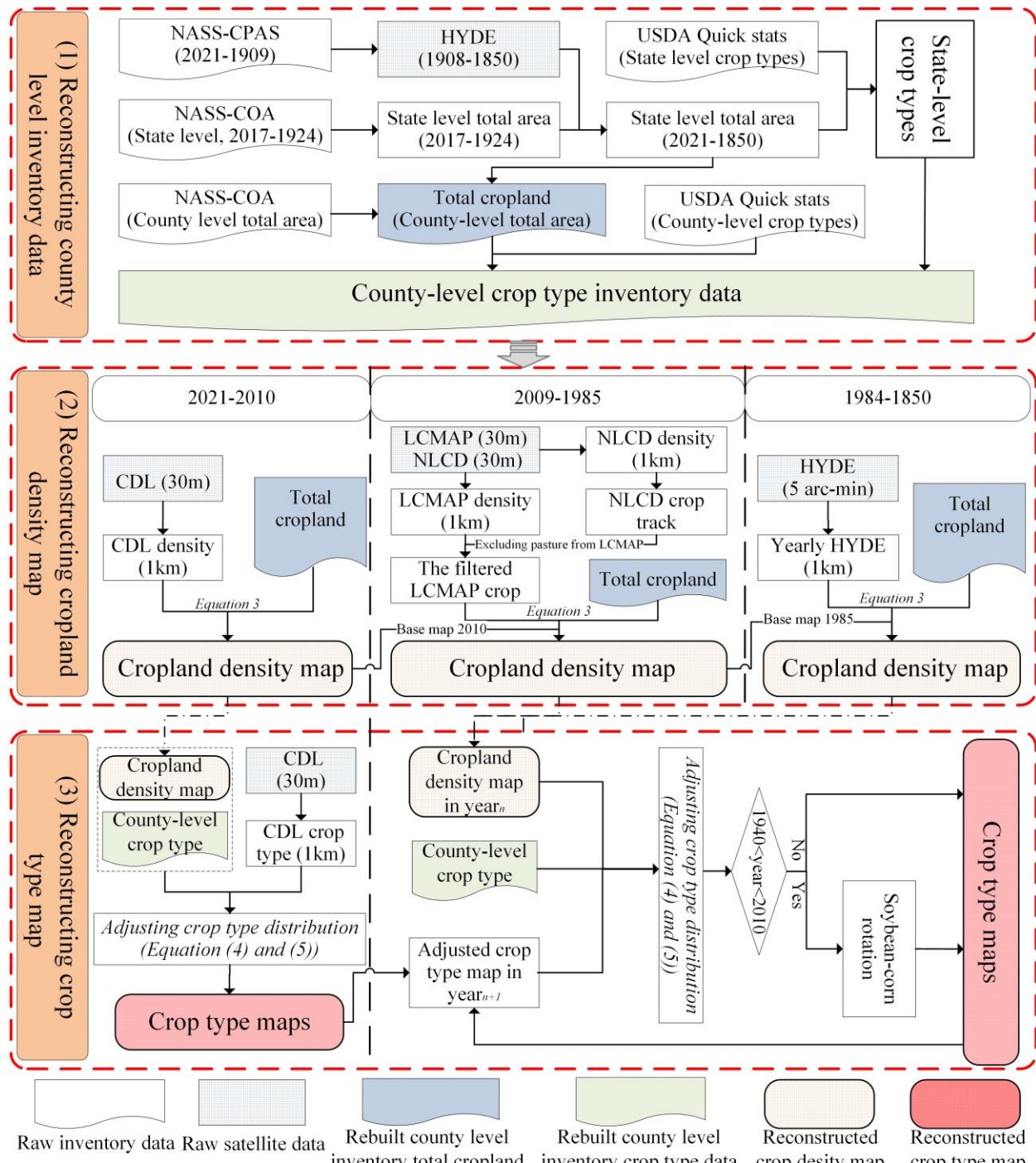


Figure 1. The methodology flow chart. Three boxes with red dashed lines correspond to Section 2.2, 2.3, and 2.4,
respectively. The county-level total and crop-specific cropland area generated in the box (1) are fed into box (2) and
box (3) to reconstruct cropland density and crop type maps, respectively. (NASS-CPAS: Crop Production Annual
Summary data from Nation agricultural statistical service of USDA; NASS-COA: Census of Agriculture from Nation
agricultural statistical service of USDA; CDL: Cropland data layer; NLCD: National land cover database; LCMAP:
Land change monitoring, assessment, and projection; HYDE: History database of the global environment 3.2
(Goldewijk et al. 2017).

## 2.1 Datasets

Three inventory datasets and four gridded LCLUC datasets were used in this study (Table 1). Specifically, NASS-
CPAS (Crop Production Annual Summary data from the Nation agricultural statistical service of USDA) and NASS-
COA (Census of Agriculture from Nation agricultural statistical service of USDA) provide the total cropland area in
each state and each county. USDA-NASS Quickstat was used to track the acreage of specific crop types. These
inventory datasets were adopted to reconstruct the historical crop-specific planting area for each county from 1850 to
2021, which served as a benchmark for adjusting the spatial maps in terms of planting acreage. CDL is the most
detailed satellite-based cropland dataset for the period of 2010-2021, which has been intensively validated by ground
truths and other ancillary data with crop classification accuracies up to 90% for major crop commodities (Boryan et
al., 2011; Yu and Lu, 2018). Here, we extracted ten crop types (Table S1) from CDL. We compared the planting area
between inventory data and CDL for nine crop types across counties from 2010 to 2021 (Figure S1). For most counties
(>75%), the residuals (the inventory-based crop area minus CDL-based crop area) are less than 10 Kha for durum
wheat while they are less than 5 Kha for other crops. NLCD and LCMAP, both derived from Landsat images with a
resolution of 30m×30m, were integrated to provide the spatial information of cropland distribution from 1985 to 2009.
NLCD crop area is highly consistent with CPAS and COA, except that the crop area was significantly underestimated
in NLCD 1992 (Figure 4 in Yu and Lu, 2018), so it was excluded for reconstructing historical crop maps (Johnson,
2013). Due to its consistency in cropland area, we utilized NLCD for identifying the spatial distribution of cropland
(Homer et al., 2020). However, NLCD provides around 5-year cyclical land cover maps from 2001 to 2019 (Homer
et al., 2020). LCMAP offers annual land use data from 1985 to 2021. LCMAP adopts Anderson Level I-based legend,
grouping cropland and pasture into one category (Xian et al., 2022). In contrast, NLCD uses a Level II-based legend
where cropland and pasture are separately tracked (Xian et al., 2022) (Table S4). To generate a reliable cropland
distribution, the long-term non-crop trajectory derived from NLCD was used to exclude all grids identified as cropland
the LCMAP map (more details are presented in Supplementary Methods: (1) Preprocesses for LCMAP). For the period
of 1850-1984, although both ZCMAP and HYDE provide the cropland distribution, HYDE considers the impacts of
various environmental factors (soil suitability, temperature, and topography) on crop distribution compared with
ZCMAP (Goldewijk, 2001; Goldewijk et al., 2011; Goldewijk et al., 2017; Zumkehr and Campbell, 2013).
Consequently, HYDE (available every 10 years) was initially used to identify the cropland distribution by calculating
the fraction of cropland to the physical area for each grid. We further linearly interpolated the fraction for the missing
years between two available years to provide a potentially continuous cropland distribution (more details are presented
in (2) Linear interpolation in HYDE of Supplementary Methods). All gridded datasets were resampled to 1km. We
employed a 1km*1km window to aggregate the total cropland area from the 30m*30m map and assigned the area to
the corresponding 1km*1km grid. To resample the CDL crop type map from 30m to 1km, the crop type in each 1km
by 1km pixel was assigned to the dominant crop type with the largest fraction of land area within the 1km*1km
window. Conversely, the cropland percentage in each 5 arc-min grid is interpolated to 1km*1km grid cells with an
assumption that cropland percentage is evenly distributed within the 5 arc-min by 5 arc-min window.

142               Table 1. The gridded and inventory dataset sources.

| Data variables (period, resolution) | Properties | Adjustment |
|---|---|---|
| CDL (2010-2021, 30m) | The most detailed crop type maps. Providing info of crop type and distribution. | Resampled to 1km and reclassified into ten crop types (nine major crop types and one type of "others"). |
| LCMAP (1985-2021, 30m) | Anderson Level I-based legend classification including eight primary land types (Xian et al., 2022). The cropland includes cropland and pasture. | Filtering pasture from cropland based on NLCD crop trajectory. |
| NLCD (2001-2019, 3-5 years intervals, 30m) | Anderson Level II-based legend including 20 land cover classes (Xian et al., 2022). | Providing cropland distribution. |
| HYDE 3.2 (1600-2017, 5arc-min) | Including cropland, grazing land, pasture, irrigated rice, etc. Providing cropland distribution. | Linear interpolation in missing years (1850-1985) (Equation S2). |
| NASS-CPAS (1909-2021) | State-level total planting area of major principal crops*. | Gap-filling in missing years (Section 2.2). |
| NASS-COA (1924-2017, 4-5 years intervals) | State and county-level total cropland area of harvest, failure, and fallow crops. | Gap-filling in missing years (Section 2.2). |
| USDA-NASS Quickstat (1866-2021) | State and county level crop-specific planting and harvesting area. Including corn, soybean, winter wheat, spring wheat, durum wheat, cotton, sorghum, barley, rice, and all other crop types. | Gap-filling in missing years (Section 2.2). |

* Principal crops refer to grains, hay, oilseeds, cotton, tobacco, sugar crops, dry beans, peas, lentils, potatoes, and
miscellaneous crops.
**2.2 Reconstructing crop acreage history at the county level**
By integrating and gap-filling multiple inventory and gridded datasets, we reconstructed the county-level time
series of planting area and the planting area for nine major crop types and other crops from 1850 to 2021. Our
reconstruction process was initiated with the development of crop-specific planting areas at the state level. NASS-
CPAS reports the annual total planting area of major crops for each state from 1909 to 2021. However, some minor
crop types, such as vegetables and fruits, are excluded. USDA-COA provides the total areas of crop harvest, failure,
and fallow for each state from 1925 to 2017 with 4~5-year intervals. We computed the difference between these two
datasets for available years and linearly interpolated unavailable years during 1909-2021. The difference was assumed
to be the planting area of those minor crops. The interpolated difference was then added back to NASS-CPAS to
generate the annual state-level total crop planting area of all crops from 1909 to 2021. We used the interannual
variations of arable land of each state extracted from HYDE to extrapolate the total planting area from 1908 to 1850
(Equation 1). To identify the planting acreage change for nine major crop types, we obtained the state-level crop-
specific harvesting and planting area from USDA-NASS Quickstat. The available harvesting and planting areas vary
among crop types and states, for which the harvesting areas usually have earlier-year reports than those of planting
areas (Table S2). The harvesting area is highly correlated to planting area in terms of interannual variation. We
calculated the ratio of planting area to harvesting area for the earliest available year of planting area. We then converted
the harvesting areas to planting areas by timing the ratio with the harvesting areas to extend the planting areas to an
earlier period. For the period that the harvesting areas are unavailable, we interpolated the planting area from 1850 to
2021 based on the total planting area generated above as a referenced trend. Equation 1 was used when only the
beginning or the ending year of the period is available, while Equation 2 was used when both beginning and ending
years are available. The planting area of "others" was obtained by calculating the difference between the total planting
area and the summation of planting area of 9 major crops.
We adopted the same approach as for the state-level planting area generated above to obtain the county-level total
planting area and the planting area of 9 major crop types and "others". USDA-COA reports the total county cropland
area from 1925 to 2017 with 4~5-year intervals.  We gap-filled the total county planting area from 1850 to 2021 by
using state total planting area as a referenced trend (using Equation 1 for gap-filling in cases where only beginning or
ending year is available and Equation 2 in cases where both beginning and ending years are known). Similar to the
state-level crop-specific planting area, we converted the harvesting areas to planting areas of nine major crops in each
county from USDA-NASS Quickstat, with varied availability (Table S1). For the period when harvesting areas are
unavailable, we gap-filled the planting areas of each crop during 1850-2021 based on the state-level crop-specific
planting area generated above as a referenced trend (Equation 1 and 2). The planting area of all other crops ("others")
in each county was estimated by calculating the difference between the total cropland area and the total area of 9 major
crops.
$Raw\ data_{i+k} = \frac{Referenced\ trend_{i+k}}{Referenced\ trend_i} \times Raw\ data_i,$      (1)
$Raw\ data_{i+k} = \frac{Referenced\ trend_{i+k} \times Raw\ data_i}{Referenced\ trend_i} \times \frac{k-i}{j-i} + \frac{Referenced\ trend_{i+k} \times Raw\ data_j}{Referenced\ trend_j} \times \frac{j-k}{j-i},$      (2)
Where $Raw\ data$ is the raw data that contains missing values, $Referenced\ trend$ is the complete data from
which the interannual variations that raw data can refer to, $i$ and $j$ are the beginning and ending year of the gap, $i + k$
is the $k$th missing year.
**2.3 Spatializing county-level cropland density**
By incorporating the county-level inventory (Section 2.2) and gridded cropland products, we reconstructed annual
cropland density maps with 1 km by 1 km resolution to represent the area and distribution of cultivated land in the
conterminous US from 1850 to 2021. This process was divided into three periods: 2010-2021 (P2010), 1985-2009
(P1985), and 1850-1984 (P1850). CDL, LCMAP, and HYDE were used to provide the potential cropland distribution
in P2010, P1985, and P1850, respectively. For the initial density maps in P2010 and P1985, we used a 1 km window
to count cropland fraction in each grid resampled from the raw CDL and LMCAP (30m×30m), respectively, while
initial annual density maps in P1850 were resampled and linear interpolated from the HYDE maps. The pixel value
in the resampled density map, representing the proportion of the cultivated land over the total pixel area, was further
corrected based on the reconstructed county-level inventory data (Equation 3).
Specifically, when the total cropland area in a county from the initial density map is larger than that of the
inventory area, the extra area from all grid cells in the initial map would be deducted to keep consistent with the
magnitude of the inventory data; On the contrary, if the cropland area was less than the inventory data, the inadequate
area would be added to all pixels (Yu and Lu 2018). If the fraction in a grid is reduced below zero, the cropland
fraction in that grid is assigned to zero and the remaining difference area between the map and the inventory data is
subtracted from other grids. Conversely, if the fraction in a grid increases above one (100%), then the value in that
grid is assigned to one, and the remaining area will be added to other grids.
$$AdjPixel_k = Pixel_k + \frac{(inv - \sum_1^n Pixel_k)}{n},\tag{3}$$
Where $n$ is the total number of valid cropland pixels in a county; $k$ is the pixel ID in that county, which is from 1
to $n$; $inv$ is the inventory crop area in that county; $Pixel_k$ is the initial cropland density in pixel $k$; $AdjPixel_k$ is the
adjusted cropland density in pixel $k$.
To eliminate the gap between CDL and LCMAP, we used the adjusted CDL 2010 density map as a baseline map
to retrieve the cropland density maps during 1985-2009 by adopting the year-to-year gridded changes from the
resampled LCMAP maps. Taking the year 2009 as an example, the interannual difference in each grid between
LCMAP 2009 and 2010 was applied to the adjusted CDL 2010 to generate the potential crop density map in year 2009.
Then, the potential density map was further corrected by the inventory data through Equation 3. Following the same
rule, the difference between the interpolated HYDE 1985 and 1984 was applied to the adjusted LCMAP 1985 to
retrieve the density maps in P1850.
**2.4 Spatializing county-level crop type map**
Based on the reconstructed county-level crop type inventory data (Section 2.2), corrected cropland density maps
(Section 2.3), and CDL, spatializing annual crop type maps was divided into two periods: 2010-2021 (P1) and 1850-
2009 (P2). For P1, the raw 30m resolution CDL crop type maps were resampled to 1 km to provide the potential crop
type distribution. In this process, we assigned the resampled grid to a type with the biggest percentage in a 1 km
window. By integrating resampled crop type maps and reconstructed cropland density maps, we counted the total area
for each type at the county level, and identified the crop types whose area is greater than the corresponding inventory
record. We further converted the surplus pixels from these types to other types whose area is less than inventory data
(Equation 4 and 5). In particular, to avoid a grid planted by a fixed type for a long time, the surplus pixels are randomly
selected for the conversion across different crop types. For P2, we assumed that the crop type pattern in two
consecutive years wouldn't change significantly, and used the rebuilt crop type map in $year_{i+1}$ to provide the potential
crop type distribution in $year_i$. Then, we followed the same rule in P1 to reconstruct the crop type map in $year_i$.
$AdjType_j = inv_j - \sum_1^n (AdjPixel_{j_k}),$                                                                (4)
Where $j$ is the crop type ID ranging from 1 to 10, which is identified from the initial crop type map; $n$ is the
number of total valid pixels in crop type $j$; $k$ is the pixel ID of crop type $j$ ranging from 1 to $n$ identified from the
initial crop type map; $inv_j$ is the inventory area of type $j$; $AdjPixel_{jk}$ is the adjusted cropland percentage in pixel $k$;
$AdjType_j$ is the crop area converted to other types; For $year_i$ between 2010 and 2021, the initial crop type map is
resampled from CDL; For $year_i$ from 1850 to 2009, crop type map is the adjusted crop type map in $year_{i+1}$.
$\begin{cases} Converting\ the\ area\ of\ AdjType_j\ from\ type\ j\ to\ other\ types, if\ AdjType_j < 0; \\ Converting\ the\ area\ of\ AdjType_j\ from\ other\ types\ to\ type\ j, if\ AdjType_j > 0; \end{cases}$                   (5)
Considering the dominant crop rotation type in US, soybean and corn rotation, we simulated the corn-soybean
rotation by randomly switching a certain area between corn and soybean according to the rotation rate. The crop
rotation information from 1996 to 2010 at state level was documented by the "Tailored Reports: Crop Production
Practices"      of      USDA's      Agricultural      Resource      Management      Survey      (ARMS)
(https://data.ers.usda.gov/reports.aspx?ID=17883). The rotation rate was calculated as the ratio of the sum of corn-
soybean and soybean-corn acreage to the total area of corn and soybean. We found that the rotation rate in each state
kept relatively stable in the ARMS-available years, and assumed that the rotation rate in the missing years is the same
as the mean rate from available years (Table S3), which is further applied to corresponding counties. Because soybean
was rarely planted in the Corn Belt before 1940 (Yu et al., 2018), we only considered the corn-soybean rotation during
the period 1940-2009 in 17 states (Table S3) (Padgitt et al., 1990).

## 2.5 Evaluation method

Here, we adopted multiple indexes to evaluate the crop area discrepancy between the reconstructed maps and
inventory data at various scales. At the county level, we utilized the residual ($resd_{ij}$) and relative residual
($relative\_resd_{ij}$) to describe the crop area difference and relative difference between the rebuilt maps and the
inventory data (Equation 6 and 7). In addition, at the national scale, the Root Mean Squared Error (*RMSE*) and R-
squared ($R^2$) are used to assess the crop area consistency between the crop maps and the inventory data.
$resd_{ij} = inv_{ij} - map_{ij},$                                                                             (6)
$relative\_resd_{ij} = (inv_{ij} - map_{ij}) * 100/inv_{ij},$                                                  (7)
Where, $inv_{ij}$ and $map_{ij}$ are the crop area derived from the inventory data and the rebuilt maps at year $i$ and in
county $j$, respectively. $resd_{ij}$ and $relative\_resd_{ij}$ are the residue and relative residue at year $i$ and in county $j$,
respectively.

### 2.6 Cropping diversity analysis

Cropping diversity has been identified as a potential factor affecting crop yield (Renard and Tilman, 2019; Driscoll et al., 2022). Here, we adopted a true diversity index proposed by Jost (2006) to analyze the US crop diversity pattern. The true diversity ($D$) quantifies the effective number of crop species (Equation 6), where a given $D$ value is equivalent to the number of crop species with an equal area in a certain space. $D$ is calculated as the exponent of Shannon diversity index ($H$).

$$D = \exp\left(-\sum_{j=1}^{n}(P_j * lnP_j)\right) = \exp(H), \tag{8}$$

Where, $P_j$ is the proportion of the cropland area occupied by crop type j over the total cropland area, and $n$ is the number of crop species. In this study, the diversity calculated involves ten crop types, including nine major crop types and a category of "others".

### 3 Result

### 3.1 Validation of the data products

In this study, we adopted the inventory data to refine the gridded map, recognizing that achieving exact alignment for each crop type within each county might be challenging due to constraints related to the limited cropland area available for allocation. Here, we examined the crop-specific area alignment between the inventory data and our map products at multiple scales. We compared the annual crop type-specific acreage extracted from our maps with the raw inventory data at county level in 1920, 1960, 2000, and 2020 (Figure S2). The county-level acreages derived from our products and inventory data are close to the 1:1 line, with $R^2$ exceeding 0.95 and $RMSE < 1$ Kha for all the major crop types except for winter wheat ($R^2 = 0.98$, $RMSE = 2.79$ Kha) and cotton ($R^2 = 0.95$, $RMSE = 3.97$ Kha). Although winter wheat and cotton present a relatively greater $RMSE$, the counties with crop area bias greater than 10% only account for 9.7% and 6.1% of total winter wheat- and cotton-planting counties in the selected four years, respectively. We further examined the time-series residual between the inventory data and maps (Figure 2 and S3). It is evident that the residuals (the inventory-based crop area minus the rebuilt-map-based crop area (Equation 7)) are generally smaller than 0.2 Kha for the majority counties (>75%) across all years for nine crop types. Relatively greater residuals are observed in spring wheat, durum wheat, and rice before 1875 (Figure 2d, g, and i), which might be attributed to the marginal area of these three crops during the early years. Similarly, the relative errors (the ratio of residual to the inventory crop area (Equation 8)) in most counties remain within ±2% for different crops, except for spring wheat, durum wheat, and rice before 1875 (Figure S3d, g, and i). We also checked the consistency in national crop-specific acreage between our maps and the inventory data during 1850-2021 (Figure S4). The results show that the map products match well with the inventory data ($R^2$ close to 1 and $RMSE < 0.3$ Mha for all crop types), indicating that the developed maps are highly consistent with the inventory data at national scale. The multiple-scale validations demonstrate that the developed dataset has the strong capacity to capture the interannual crop-specific area variation.


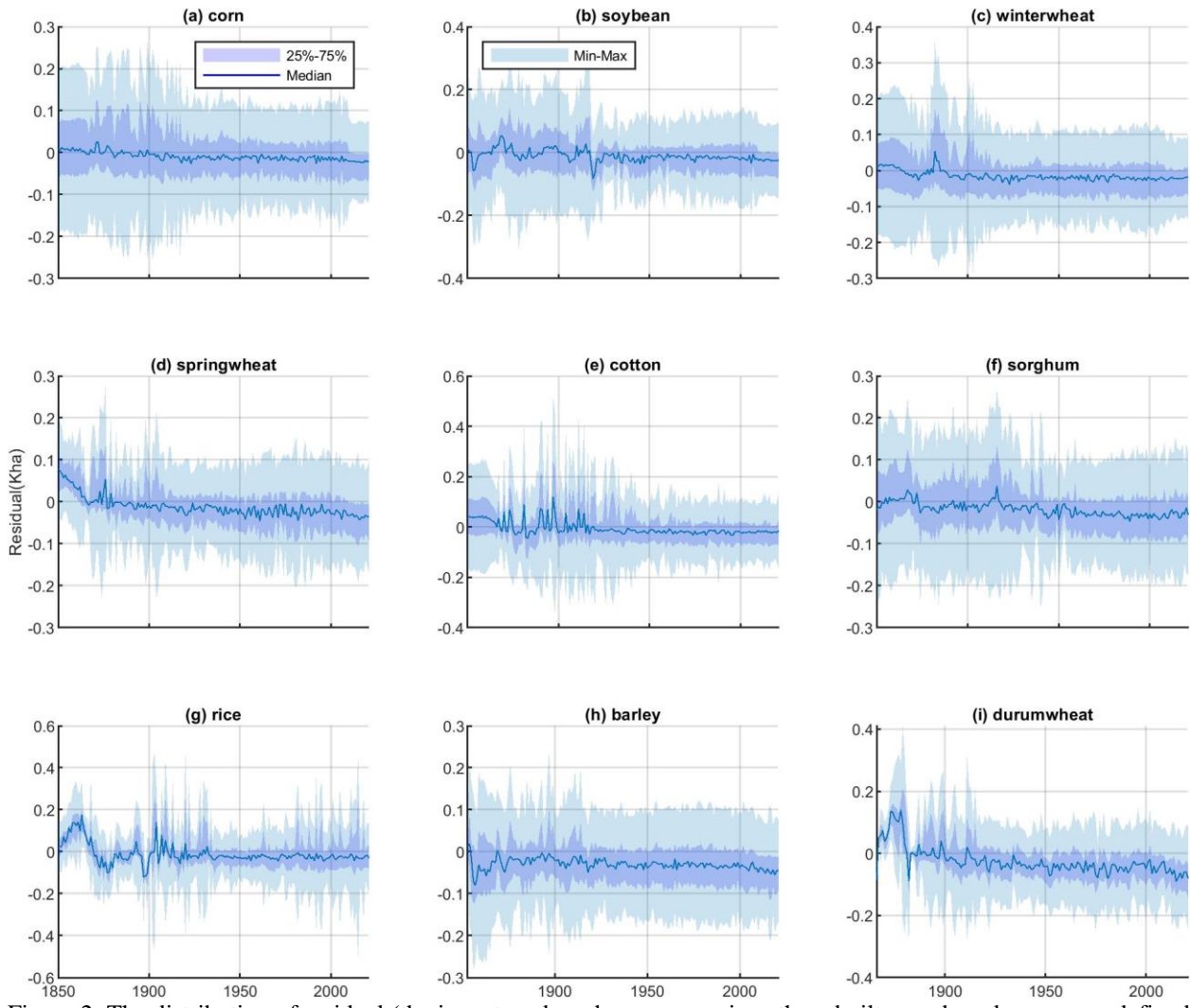

Figure 2. The distribution of residual (the inventory-based crop area minus the rebuilt-map-based crop area, defined
by Equation 6) between the rebuilt inventory and maps from 1850 to 2021 (Kha is a thousand hectares). In each year,
"Min-Max", "Median", and "25%-75%" reflects the extent of residual from all counties at levels of "minimum value
to maximum value", "50th percentile", and "25th percentile to 75th percentile", respectively, which are corresponding
to five percentiles in a box plot.

290       We examined the historical changes in cropland area among various crop types in the US from 1850 to 2021
(Figure 3). In general, the US cropland expanded rapidly from 21.66 Mha in 1850 to 149.28 Mha in 1919, followed
by a wide fluctuation ranging from 134.78 Mha to 161.80 Mha until 1990, and then kept relatively stable around
140.00 Mha until 2021. Corn was the dominant crop in the US, accounting for more than 20% of the national total
cropland area throughout the study period. Temporally, it rose sharply from 7.47 Mha in 1850 to 50.47 Mha in 1917,
followed by a continuous drop to 26.26 Mha until 1962, and slowly increased to 37.75 Mha during 1962-2021.
Soybean soared significantly from 4.35 Mha in the 1940s to 35.25 Mha in 2021, becoming the second most extensive
crop type in the US. Winter wheat constantly increased from 3.25 Mha in 1850 to 26.43 Mha in 1981 and then dropped
to 12.88 Mha in 2021, while spring wheat fluctuated dramatically after it plateaued at 8.28 Mha in 1933. Barley and
sorghum climbed to peaks of around 8 Mha in 1940s and 11 Mha in 1950s, and then dropped to about 1 Mha and 3

Mha by 2021, respectively. Besides, cotton and durum wheat both reached their peaks before the 1930s and then fell to a relatively stable level. Throughout the study period, the total US cropland increased by 118 Mha, predominantly driven by corn (30 Mha), soybean (35 Mha), and others (31 Mha). The remaining row crops shared about 18% of this increase, including winter wheat (9.6 Mha), spring wheat (4.5 Mha), sorghum (2.8 Mha), cotton (2.7 Mha), and rice (1 Mha).

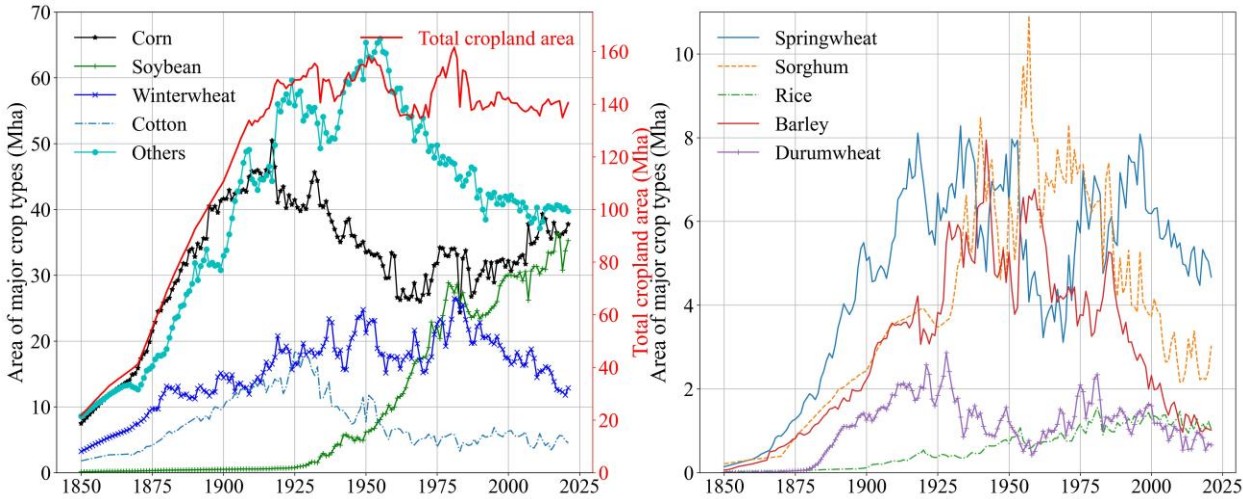

Figure 3. Annual area of major crop types and total US cropland area from 1850 to 2021.

## 3.2 Dynamics of cropland distribution

The spatial patterns of cropland density and crop type are presented in Figure 4. Generally, the hotspots of cropland are concentrated in the Midwest and Great Plains regions (the spatial pattern of US subregions showed in Figure 5(2-a)), starting from 1950, where large crop field sizes were likely to occur (Yan and Roy, 2016). The results show that the cropland was mainly distributed in the eastern region of the US in 1850 with a low distribution percentage (< 40%) (Figure 4(a)). Then, the cropland density enhanced substantially (40%-80%) in 1900 (Figure 4(b)). Meanwhile, a large area of the Great Plains was cultivated to plant corn and spring wheat in the Northern Great Plains and winter wheat in the Southern Great Plains during 1850-1900 (Figure 4(f)). From 1900 to 1950, the cropland fraction was continuously elevated (>60%) (Figure 4(c)), especially in the Midwest and the Great Plains. During 1950-2021, spring wheat expanded westward to Montana (Figure 4(h)), enhancing the cropland fraction in the Northern Great Plains. Moreover, the category of "others" substantially substituted corn, winter wheat, and cotton in the Southeast of US, and lowered the cropland density in this region (Figure 4(d)). It was noted that the soybean increased tremendously since 1950 in the Midwest, the Dakotas, and the rice belt, replacing parts of spring wheat, winter wheat, barley, and rice in these regions. Overall, the hotspots of US cropland have shifted from the Eastern US to the Midwest and the Great Plains with the increasing cropland percentage over the past 170 years.

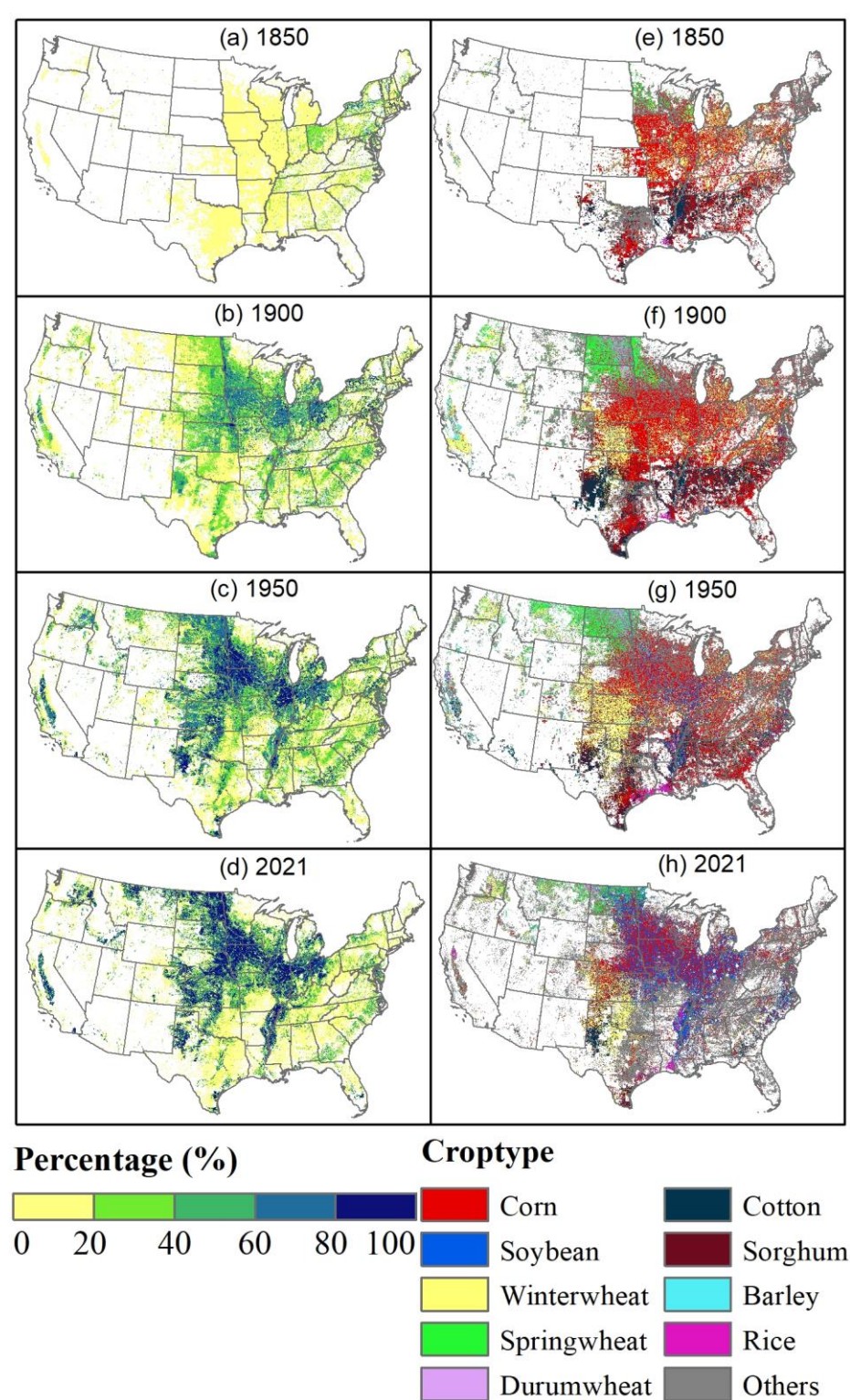

Figure 4. The spatial patterns of cropland percentage (a-d) and crop type (e-h) at 1 km by 1km resolution in 1850,
1900, 1950, and 2021. The color bar of "Percentage" indicates the percentage of planting area to the grid area. "Others"
represents the remaining crop types.
Furthermore, the spatiotemporal patterns of each major crop type were examined in this study to present a
systematic understanding of the US cropland extent and type changes (Figure 5, Figure S5 and S6). Specifically, corn

was mainly planted in the east in 1850, with a low cropland fraction (<40%) (Figure 5(1-a)). Then, it gradually expanded to the Great Plains, and the total area increased by 43 Mha from 1850 to 1917. Meanwhile, the hotspots of corn planting areas shifted to the Midwest, the southeast of the Northern Great Plains, and the northeast of the Southern Great Plains (Figure 5(1-b)). From 1917 to 1962, the spatial extent of corn had shrunk in South Dakota, Nebraska, Kansas, and the Southeast, with a total area decrease of 24.21 Mha (Figure 5(1-c)). Although the Southeast experienced a large decline in corn acreage during 1962-2021, the planting density of corn significantly increased in the Midwest and the southeast of the Northern Great Plains, resulting in the corn area peaking at 37.75 Mha in 2021 (Figure 5(1-d)).

Temporally, soybean was rarely cultivated in the US from 1850 to 1900 with a total area less than 1 Mha (Figure 5 (2-a and 2-b)). During 1900-1940, the planting area of soybean had a small expansion in the Midwest, with a total area rising to 4.35 Mha (Figure 5(2-c)). But then, it had a dramatic expansion from 1940 to 2021 to the Midwest, Southeast, and the east of Northern Great Plains, with the total soybean area increasing to 35.25 Mha (Figure 5(2-b)).

Winter wheat was mainly located in the Midwest in 1850 with a total area of 3.25 Mha (Figure 5(3-a)). In the following five decades, it spread to the Great Plains, California, Washington, and Oregon, with the total area increasing to 14.45 Mha in 1900 (Figure 5(3-b)). From 1900 to 1981, although its spatial extent had shrunk in Midwest, it expanded significantly in the Southern Great Plains, the Southeast, and Montana (Figure 5(3-c)). Meanwhile, the cropland density also enhanced in this period. These changes led to the planting area of winter wheat reaching the peak of 26.43 Mha in 1981. However, during 1981-2021, a large area of winter wheat was replaced by other crop types or other land use types in the Midwest, Southeast, Montana, Washington, and California (Figure 5(3-d)), which reduced the total area of winter wheat to 12.88 Mha in 2021.

Cotton was mainly distributed in the Southeast in 1850 with a low density (Figure S5(1-a)). It sharply expanded to the Southern Great Plains and California with the increased density during 1850-1925 (Figure S5(1-b)), and the total area of cotton increased by 16.53 Mha in this period. But the period of 1925-2021 was characterized by a huge contraction of cotton area in the Southeast and Southern Great Plains, with a total area declining to 4.50 Mha (Figure S5(1-c and 1-d)).

For spring wheat, there was a significant expansion from Montana and Wisconsin to the Midwest and Northwest during 1850-1933, resulting in a total area increase to 8.28 Mha (Figure S5 (2-a) and (2-b)). But the distribution of spring wheat had largely shrunk in the Midwest and Northwest from 1933 to 1969 (Figure S5 (2-b) and (2-c)), resulting in the area decreasing to 3.11 Mha. In recent decades, it mainly centered in the northern part of the Northern Great Plains with the enhanced density in each grid, and its total area increased to 4.67 Mha in 2021 (Figure S5 (2-d)).

Sorghum consistently expanded in the Southern Great Plains from 1850 to 1957, with its total area increasing by 10.70 Mha (Figure S6 (1-a to 1-c)). However, there was a subsequent area decline thereafter, leaving the total at 3.03 Mha in 2021 (Figure S6 (1-d)). Similarly, barley experienced a continuous expansion in the Midwest, Great Plains, Northeast, California, and Colorado, with the total area rising from 0.06 Mha in 1850 to 7.94 Mha in 1942 (Figure S6 (2-b to 2-c)). However, between 1942 and 2021, the distribution of barley had a dramatic contraction across the entire US and shrank to 1.02 Mha in 2021, with a small extent in the Northern Great Plains (Figure S6 (2-d)).

Compared with other major crop types, both the distribution of durum wheat and rice only occupied a small area
of the US over the entire study period (<3 Mha). Specifically, durum wheat underwent significant expansion in North
Dakota and South Dakota from 1850 to 1928 (Figure S5 (3-a and 3-b)), reaching a peak area of 2.86 Mha in 1928.
Subsequently, it contracted to the eastern part of North Dakota during 1928-1958, with a total area declining to 0.42
Mha (Figure S5 (3-c)). From 1958 to 2021, its planting area shifted to the junction of North Dakota and Montana
(Figure S5 (3-d)). Rice consistently expanded in Arkansas, Louisiana, Mississippi, and Texas from 1850 to 1981,
resulting in a total area increase of 1.55 Mha (Figure S6 (3-a to 3-c)). This expansion gradually formed the current
rice belt pattern, followed by a small shrinkage (0.52 Mha) in these regions between 1981 and 2021 (Figure S6 (3-d)).
The category of "others" includes various minor crop types such as peanuts, oats, alfalfa, etc., collectively accounting
for 27%~43% of the total US cropland area and distributing across the entire US (Figure S5).

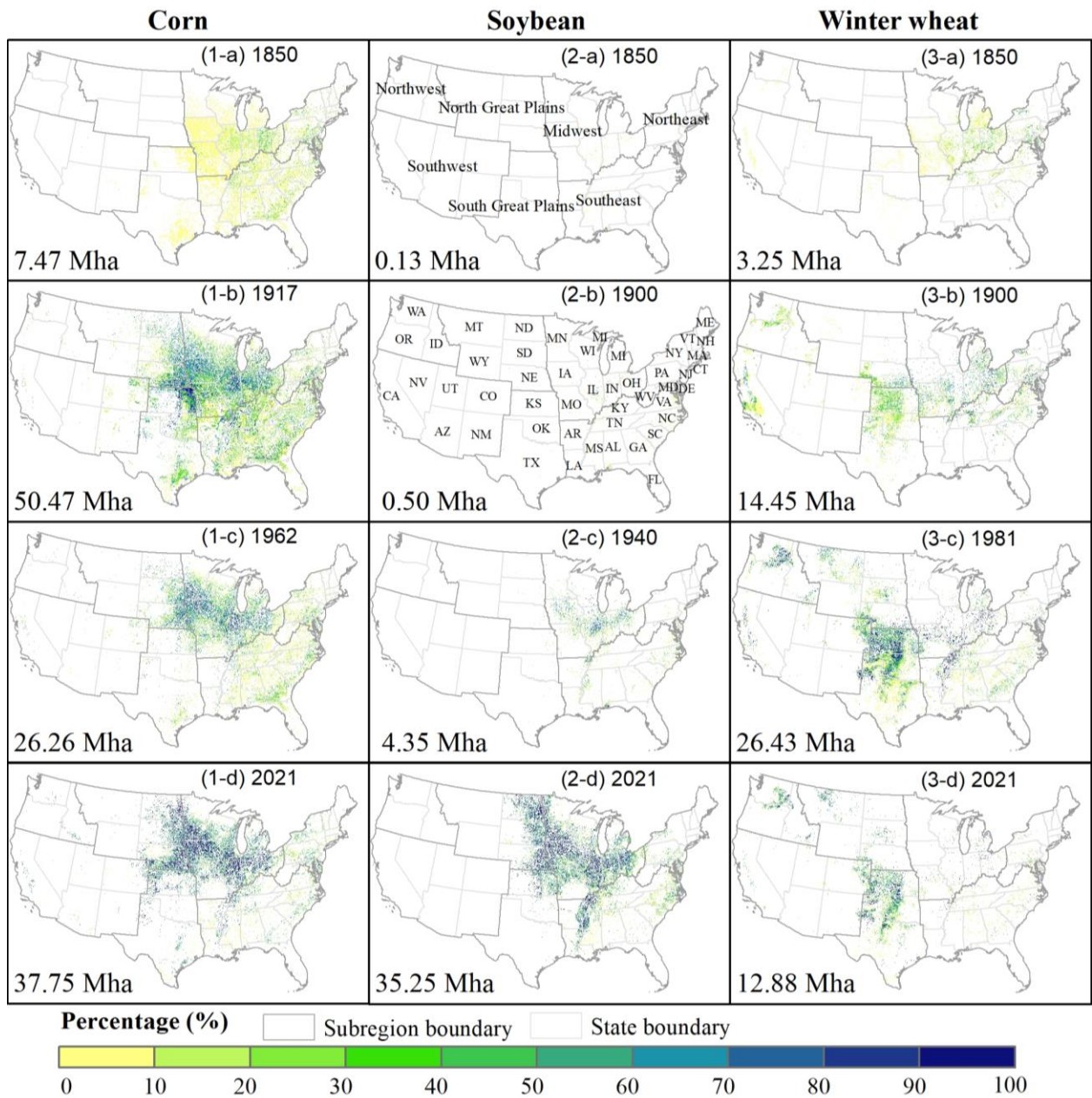

Figure 5. The spatial density pattern of corn, soybean, and winter wheat at 1km by 1km resolution in the area turning years. The first, second, and third columns are the density pattern of corn, soybean, and winter wheat, respectively. The total planting area for each crop type is presented in the bottom left of each subfigure. The color bar at the bottom indicates the percentage of planting area to the total grid area.

**3.3 Changes in cropping diversity over time**

Here, the value of true diversity (*D*) is interpreted as the number of crop species with an equal area in a certain space (L Jost, 2006; Hijmans et al., 2016), so a higher D value reflects more crop types, or more even distribution, or both. As shown in Figure 6, the US cropping system diversity had undergone dramatic change over time, with a sharp increase from 1850 to 1963 and a significant decline in the recent 60 years. Among different regions, the Southwest, Northern Great Plains, Southern Great Plains, and Southeast had a higher cropping system diversity than the remaining

regions. Specifically, the diversity in Southwest, Southern Great Plains, and Northern Great Plains presented a similar change during 1850s-1940s, with a drop from 1850s to 1880s followed by an obvious increase to 1940s (Figure 6 (b)). Starting from 1940s, the diversity in Northern Great Plains peaked around 1990s and then constantly decreased to 2021, while Southern Great Plain's diversity presented an opposite trend in this period. Meanwhile, Southwest witnessed a continuous decline in crop diversity from 1940s to now. The Southeast kept its diversity stable during 1850s-1930s and then experienced a significant increase from 1940s to 2000s. However, in the recent 20 years, the diversity in Southeast dropped sharply. The diversity in Northeast showed an increase trend across the entire study period. Northwest's crop diversity fluctuated between 2.5 and 3 from 1850s to 1970s and then had a continuous increase to now. Midwest's crop diversity kept relatively stable during 1850s-1920s. After increasing to its peak between 1920s and 1930s, it kept stable from 1930s to 1980s, followed by a dramatic decrease to 2021.

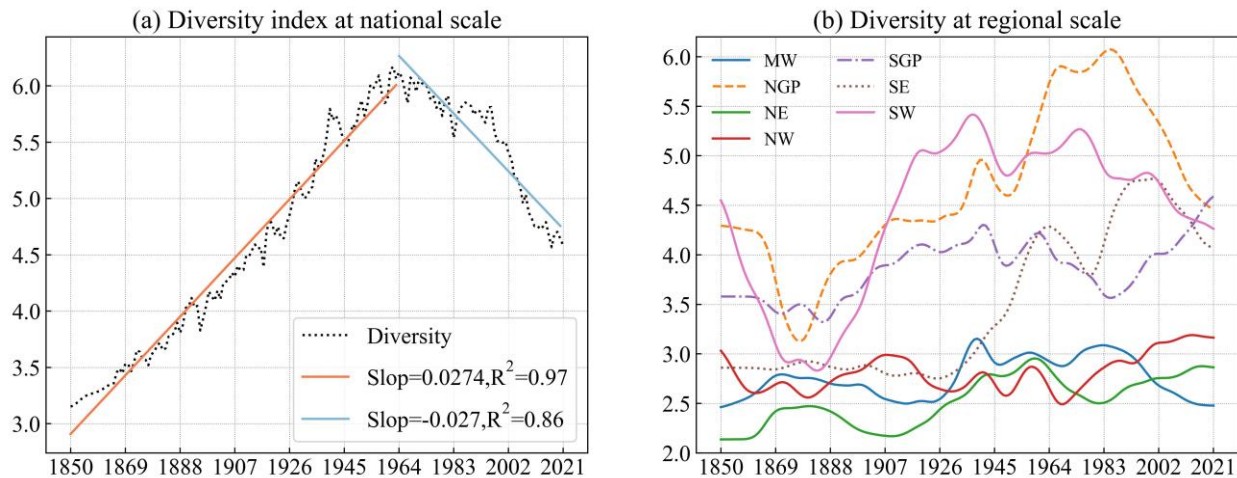

Figure 6. The temporal trend of diversity value in US (a) and seven regions (b). NW, SW, NGP, SGP, MW, SE, and NE are the abbreviation of Northwest, Southwest, Northern Great Plains, Southern Great Plains, Midwest, Southeast, and Northeast, respectively. The spatial map of seven regions is presented in Figure 5 (2-b). To get a better visual pattern, the trends of seven regions in (b) were smoothed by the gaussian function. The diversity value is calculated based on the reconstructed inventory data.

## 4    Discussion

### 4.1  Comparison with other datasets

We systematically compared our product with previous datasets regarding the historical total cropland area (Figure 7) and their spatial patterns (Figure 8) to provide a complete reference for potential applications. By combining NASS-CPAS and NASS-COA to reconstruct state- and county-level inventory data, the US total cropland area derived from our density maps matches well with that from NASS-CPAS from 1850 to 1940 and consistently aligns with the magnitude of NASS-COA and the interannual variations of NASS-CPAS between 1940 to 2021 (Figure 7). We extracted the US total cropland area from two widely used geospatial satellite products (USDA-CDL and USGS-NLCD) in recent two decades. These two datasets demonstrate a smaller area than that of NASS-COA before 2017, but their estimation of crop area magnitude and interannual variation have demonstrated greater consistency with this study over the recent five years. Meanwhile, Yu and Lu (2018) and Li et al. (2023) all used NASS-CPAS to develop

YLMAP and CONUS, respectively, resulting in a lower US total cropland area after 1940 than this study. This is
because the NASS-CPAS only includes the cropland area of principal crops in each state, which is lower than the total
cropland area reported by NASS-COA, especially after 1940. Among the existing databases, LCMAP, HYDE, GBC,
and ZCMAP represented an upper bound of the US total cropland area. Especially for GBC, it reported the national
total crop acreage about 50% higher than the upper range of all other data products (~300 Mha vs ~200 Mha around
the 1980s in Figure 7).
The divergence among these data products is mostly caused by different cropland definitions and cropland map
generation processes. Spatially, we observed that HYDE exhibits broader cropland extent and a higher fraction of
cropland per grid than our products, particularly in regions with low-density cropland distribution, such as the
Northwest, Southeast, and Southwest (Figure 8 and Figure 9). This disparity might be attributed to the definition of
cropland in HYDE, which includes both arable land and permeant cropland (Goldewijk, 2001) while our map
exclusively accounts for crop planting area of crops. More importantly, the crop planting area of our map was
constrained based on county level inventory data. Meanwhile, HDYE spatialized the subnational level inventory data
to allocate cropland area to each grid in accordance with "cropland suitability maps" informed by dynamical social
(historical population density) and stable environmental (soil suitability, temperature, and topography) information
(Klein Goldewijk et al., 2011; Yu and Lu, 2018). As a result, greater acreage and wider extent of cropland were
estimated by HYDE and were allocated to each grid (Figure 7, Figure S8, and Figure S9). Similarly, the category of
cropland in LCMAP and ZCMAP contains crop and pasture (Zumkehr and Campbell, 2013; Xian et al., 2022), while
GBC cropland refers to arable land (Goldewijk et al., 2017; Cao et al., 2021), leading to their higher cropland area
than our result (Figure 7). Also, the grid density of ZCMAP was higher than this study in low-density regions (the
first row in Figure 9) because ZCMAP adopted an assumption that the historical spatial crop pattern kept roughly
similar to the basemap 2000, in which the fraction in each grid is higher in these regions (Ramankutty et al., 2008;
Zumkehr and Campbell, 2013). Moreover, CONUS showed a more extensive cropland distribution than our maps
(especially in the Great Plains and Southeast, Figure 8 and the third row in Figure 9). This is likely because they
produced more potential cropland grids than the county records through an artificial neural networks-based land cover
probability occurrence model (Li et al., 2023). GBC feeds population density and eight biophysical variables
(including elevation, temperature, soil water, etc.) into a random forest model to generate the cropland distribution
(Cao et al., 2021). As a result, the spatial pattern between GBC and our maps shows a high agreement at the national
scale (Figure 8). However, the cropland percentage in each grid cell of GBC is significantly higher than other maps
(Figure 8 and the second row in Figure 9), which might be related to the base map used in their study and the lack of
inventory records for limiting the total cropland area in US (Cao et al., 2021).
In terms of spatial details among these datasets, our products, YLMAP, CONUS, and GBC (1km×1km) can
provide more detailed spatial information than HYDE and ZCMPA (5 arc-min) (Figure 9). Furthermore, compared
with YLMAP, CONUS, and HYDE incorporating state-level census, our products are likely to demonstrate more
reliable cropland density heterogeneity within state (the third row in Figure 9) since we adopted county-level census
to control the total cropland area in each county. Thus, the rebuilt map is capable of capturing spatial shifts between
counties within a same state, such as cropland abandonment in some counties but expansion in others (Figure 9). This
indicates that the county inventory-derived datasets are more appropriate for subregion applications (Yang et al., 2020).
Overall, our product keeps highly consistent with the county-level inventory data and presents similar cropland
distribution to YLMAP and GBC that involves both biophysical and socioeconomic drivers to generate crop pixels.
In addition, unlike cropland involving arable land in HYDE or harvesting land in CONUS mentioned above, the
definition of cropland in our product refers to the crop-planting areas and excludes idle/fallow farm land and cropland
pasture, providing real surface information disturbed by agriculture. This improvement enhances the estimation
cropland change's effect on the environment. Therefore, the developed maps can provide a more comprehensive
cropland tracking for ecological and environmental assessment, covering both cropland distribution and crop types at
national and regional scales.

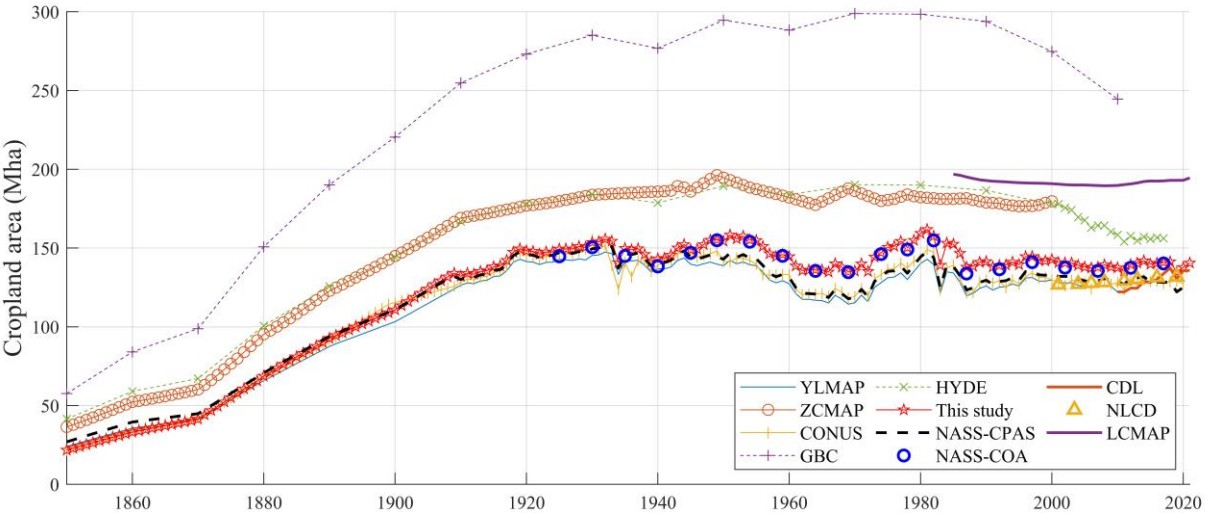

Figure 7. Comparison of the US total cropland area from different sources. CDL: Cropland data layer; NLCD: National
land cover database; LCMAP: Land change monitoring, assessment, and projection; YLMAP: the US cropland map
from Yu and Lu (2018); ZCMAP: the US cropland map from Zumkehr and Campbell (2013); CONUS: the cropland
map from Li et al.(2023); GBC: the US cropland extracted from the global cropland dataset developed by Cao et al.
(2021); HYDE: History database of the global environment 3.2 (Goldewijk et al., 2017); NASS-CPAS: the Crop
Production Annual Summary data from Nation agricultural statistical service of USDA; NASS-COA: the Census of
Agriculture from Nation agricultural statistical service of USDA. In particular, YLMAP, ZCMAP, CONUS, and GBC
are not used in this study.

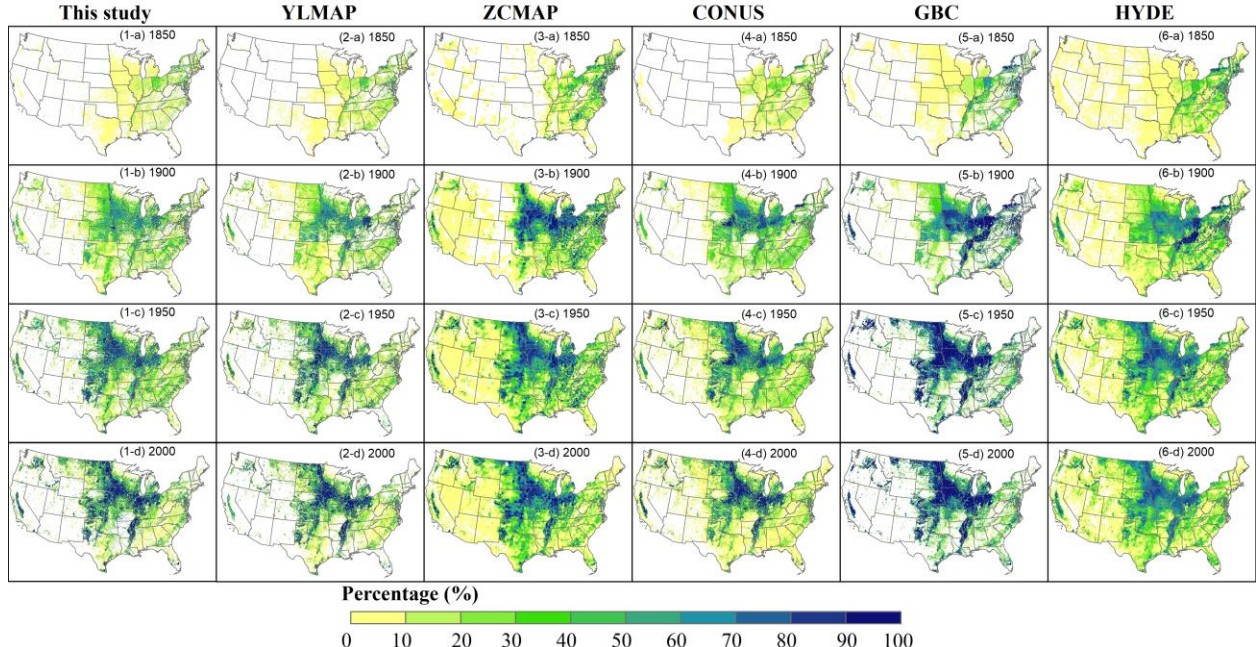

Figure 8. The spatial patterns of cropland from different datasets in selected years of 1850, 1900, 1950, and 2000. YLMAP (1km): the US cropland map from Yu and Lu (2018); ZCMAP (5 arc-min): the US cropland map from Zumkehr and Campbell (2013); CONUS (1km): the cropland map from Li et al. (2023); GBC (1km): the US cropland extracted from the global cropland dataset developed by Cao et al. (2021); HYDE (5 arc-min): History database of the global environment 3.2 (Goldewijk et al. 2017).

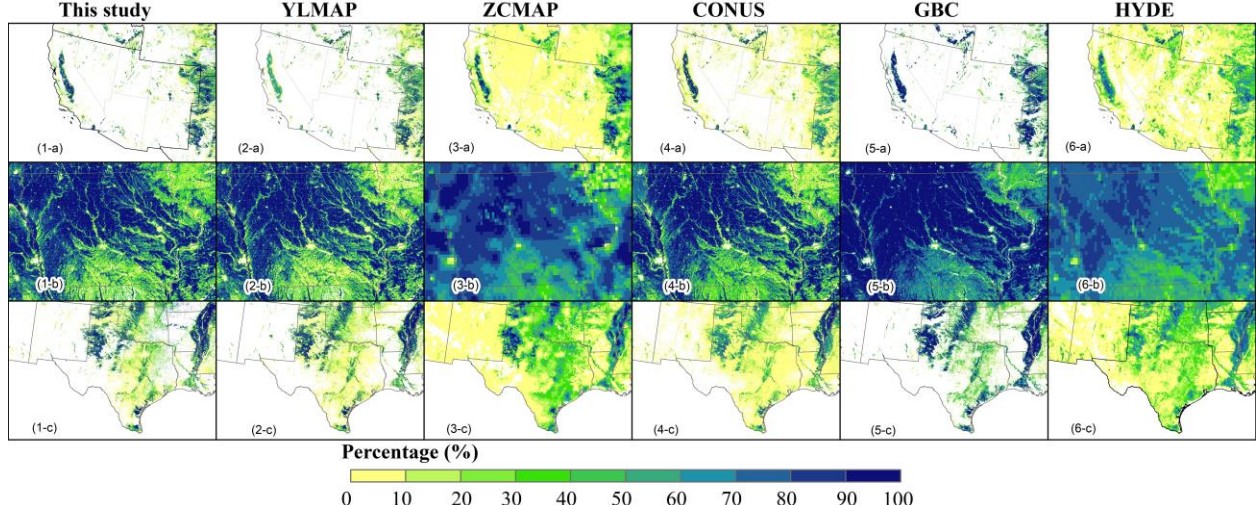

Figure 9. The detailed spatial pattern from different datasets in the year 2000. YLMAP (1km): the US cropland map from Yu and Lu (2018); ZCMAP (5 arc-min): the US cropland map from Zumkehr and Campbell (2013); CONUS (1km): the cropland map from Li et al. (2023); GBC (1km): the US cropland extracted from the global cropland dataset developed by Cao et al. (2021); HYDE (5 arc-min): History database of the global environment 3.2 (Goldewijk et al. 2017). The spatial extent in each row from (a) to (c) is Southwest, Iowa, and Texas, respectively.

## 4.2 The drivers for US cropland change

Between 1850 and 1900, there was a notable cropland expansion toward the west (Figure 4). This was mainly driven by the Homestead Act of 1862, which provided 160 acres of land to the public for farming purposes (Anderson,

2011). Additionally, the end of the Civil War, the disbanding of armies, and the building of canals and railroads toward
the west, further contributed to the agricultural market and export, accelerating agricultural reclamation (Ramankutty
and Foley, 1999). At the same time, corn, cotton, and wheat were the dominant crop types and expanded rapidly to
the west (Figure 5 and Figure S2). From 1900 to 1950, advanced irrigation systems, industrial technology, and
mechanization further promoted agricultural development. For instance, the areas of winter wheat, sorghum, and
barley increased substantially in this period (Figure 5 and Figure S2-S3). Subsequently, the fluctuation of the market,
policy structure, and weather conditions played a dominant role in affecting the interannual variations of agricultural
areas (Spangler et al., 2020). For example, the farm crisis of 1980s resulted in a significant cropland drop. Moreover,
a series of historical acreage-reduction programs, such as the conservation adjustment act program, cropland acreage-
reduction program, and conservation reserve program, resulted in the total cropland reduction (Lubowski et al., 2006).
In the recent three decades, the total US cropland has kept relatively constant, but the crop commodities changed
significantly. Corn and soybean gradually became the predominant types due to the rising demand for corn as biofuel
and the higher market price for soybean, which pushed framers to convert other types to corn and soybean (Bigelow
and Borchers, 2017; Aguilar et al., 2015). Overall, the US cropland experienced significant growth between the 1850s
and 1920s, driven by population growth, industrialization, mechanization, and market change. It subsequently
underwent a process of stabilization after experiencing fluctuations in crop types and area.
**4.3  The implications for cropping diversity change**
In general, the US cropping diversity experienced a dramatic change throughout the entire period. From 1850 to
1963, it constantly increased (Figure 6 (a)), primarily attributed to the rising areas of all major crop types during this
stage (Figure 3). Spatially, the diversity increases in the Southwest, Southeast, and Great Plains contributed to the
overall increase in US cropping diversity (Figure 6 (b) and 10). From 1960s to 2021, the cropping diversity had a
significant decrease mainly due to the increased planting area for corn and soybean and the decreased cultivated area
for winter wheat, spring wheat, sorghum, and barley. Meanwhile, the diversity drop in the Northern Great Plains,
Southwest, Southeast, and Midwest might contribute to the US crop diversity decline (Figure 6 (b) and 10). This
finding shows a strong agreement with the results of Aguilar et al. (2015), in which the crop species diversity declined
from 1980s to 2010s in the Heartland Resource Region.
On the other hand, crop species diversity is an important component of biodiversity within a cropping system,
and a decrease in crop species diversity is often associated with a decline in overall biodiversity (Altieri, 1999). Some
researchers have pointed out that the biodiversity plays an essential role in the functioning of real-world ecosystem.
High biodiversity would increase soil fertility, mitigate the impact of pests and diseases, improve resilience to climate
change, and promote food production and nutrition security(Altieri, 1999; Duffy, 2009; Frison et al., 2011). For
example, Delphine and David's research indicated that crop species diversity could stabilize food production (Renard
and Tilman, 2019), and Emily et al. (2019) found that agricultural diversification can increase crop production. Thus,
had this significant drop in the US cropping diversity in the past six decades affected yield and ecosystem productivity?
Moreover, under more frequent climate extremes anticipated in the future, whether the decreasing cropping diversity
will affect the sustainability and resilience of the US agricultural system is an important question to answer.

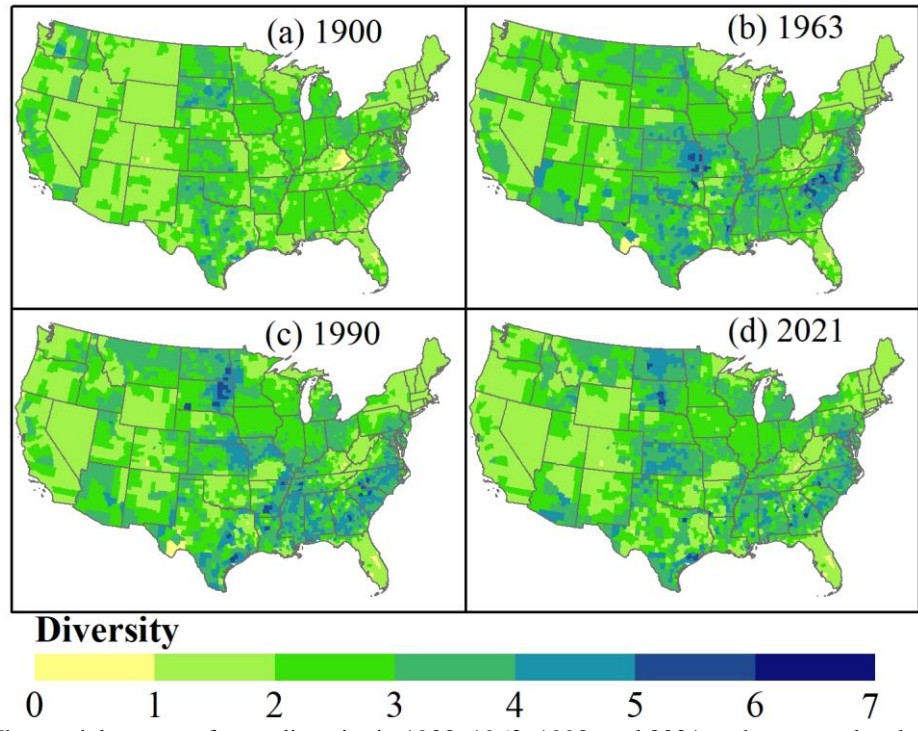

Figure 10. The spatial pattern of crop diversity in 1900, 1963, 1990, and 2021 at the county level. The diversity
520          value is calculated based on the gap-filled and multi-source harmonized inventory data in each county.

### 4.4 Uncertainty


In this study, we integrated the inventory data and the gridded LCLUC products to generate annual cropland
density and crop type maps at a resolution of 1 km×1km from 1850 to 2021. Although our data is highly consistent
with inventory data, some uncertainties remain:
(1) In the upscaling process of CDL from 30m to 1km, we assigned each pixel to a dominant crop type with the biggest
fraction of land area within the pixel. Although the area of each crop was constrained by the inventory data at the
county level, this resampling process may overlook certain crop type distributions with minor fraction within a pixel.
(2) The inventory is crucial for reconstructing historical cropland maps. Here, the rebuilt inventory data in missing
years is interpolated. Although this study is based upon our best knowledge and available, this method may not reflect
the real interannual cropland area fluctuations in the missing years.
(3) In the process of spatializing crop types, we randomly convert the cropland grids from specific types with higher
map area than inventory data to other crop types within each county. In addition, grids identified with corn-soybean
rotation were randomly selected within a county based on the corn-soybean rotation ratio, aiming to prevent a grid
cell from being consistently occupied by a single crop type over time. While the extent of the random processes varied
among counties based on the difference between intermediate map data and inventory data, it is important to note that
they may influence the temporal trajectory of grid-based crop type changes. Thus, users should exercise caution when
employing this data product for time sequencing analyses, such as crop rotation patterns (e.g., continuous corn, corn-
soybean-corn, etc.) at the pixel level.
(4) The diversity in this study mainly reflects the change in diversity among ten crop types (nine major types and one
category of "others"). It is important to note that "others" in the study is not a single crop type, but a combined category
including various minor crop types (peanuts, oats, etc.). Thus, the diversity changes quantified in this study capture
the diversity of major row crops (accounting for 70% of the national total cropland area in the 2010s) and the "others-
as-one-category" in the US over time. A more comprehensive diversity analysis involving all crop types would require
a more detailed time-series crop type record, which is currently lacking.
**5   Data availability**
The developed dataset is available at https://doi.org/10.6084/m9.figshare.22822838.v2(Ye et al., 2023). This
dataset includes annual cropland density map and crop type map with Geotiff format at 1km by 1km spatial resolution.
**6   Conclusion**
In this study, the annual cropland density and crop type map from 1850 to 2021 in the conterminous US was
developed by integrating the multi-source cross-scale inventory and gridded datasets. In general, our maps have a high
consistency with inventory data both at the national level ($R^2$>0.99, *RMSE* <0.3 Mha) and county level (the residual
less than 0.2 Kha for most counties (>75%) ). Compared with other datasets, the spatial pattern of the developed maps
matches well with YLMAP and GBC. Throughout the study period, the total US cropland increased by 118 Mha,
mainly driven by corn (30 Mha), soybean (35 Mha), and others (31 Mha). The hot spots have shifted from the East to
the Midwest and the Great Plains. Specifically, the Homestead Act of 1862 significantly contributed to the cropland
expansion toward the west, and the rising demand for biofuel and the elevated market price resulted in the dramatic
increase of corn and soybean planting areas. Meanwhile, the intensified corn and soybean substituted other crops,
leading to the decrease of the cropping diversity in the Midwest, which may further influence crop yield and co-benefit
of agroecosystem services. Additionally, there were random processes in generating crop type maps. This might bring
uncertainty to pixel-based crop type sequence detection, but the area for each crop type was well constrained by gap-
filled long-term inventory data at the county level. The county-level area control also enables the developed maps to
depict regional spatial shifts within state. Different from previous datasets, the cropland in our products refers to the
planting area of all the crops, excluding idle/fallow farm land, and cropland pasture. Hence, the cropland map provides
reliable cultivated information and reveals the surface disturbance conducted by agricultural activities, which can
improve the estimation of cropland change's impact on climate system. Overall, the developed datasets provide a
historical cropland distribution pattern, filling the data gap by providing long-term crop extent and type maps. We
envision this database could better support the US agricultural management data development with crop-specific
information, as well as improve the environmental assessment and socioeconomic analysis related to agriculture
activities.

**Acknowledgments.** This work is supported partially by NSF grant (1903722), NSF CAREER (1945036), and USDA AFRI Competitive grant (1028219). We appreciate the constructive comments and suggestions from two anonymous reviewers that have substantially improved the quality of this manuscript.

**Author contributions**: CL designed the research; SC and PY implemented the research and analyzed the results; SC, PY, and CL wrote and revised the paper.

**Competing interests.** At least one of the (co-)authors is a member of the editorial board of Earth System Science Data.

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
