# Peer review of "Annual time-series 1-km maps of crop area and types in the conterminous US (CropAT-US): Cropping diversity changes during 1850-2021"

_Earth System Science Data, 2023_

## Author Comment (AC1)

**Reviewer#1:**

This manuscript derived CONUS 1km x 1km cropland density, crop type, and crop diversity maps for 1850 to 2021, by combining several historical cropland distribution datasets. The datasets include statistics data such as USDA Quckstat (county-level plant areas of different crops), and the maps like CDL, NLCD and LCMAP generated using satellite data. The main technical approach is temporal linear interpolation to fill gaps in early years when there were no data available.

The main validation was linear regressions for derived-crop-areas vs. raw-crop-areas for 9 major crops, at county level considering four years (1920, 1960, 2000, and 2020).

In the results, the cropland density, crop type, and crop diversity maps for selected years in 1850-2021 were compared, and some patterns were found.

The findings are plausible, but the method needs clarify and justifications, and the reliability of the derived maps for was not sufficiently presented. Detailed comments are given below.

**Response**: We are grateful for the reviewer's valuable comments. We agree with the reviewer that the method and reliability of our map products were not sufficiently presented. In our revised manuscript, we improved the readability of the method section and strengthened the validation of spatial maps.

Major Comments:

*1. The validation should show regression residuals across different years in the study period, especially for the early years when the gap-fillings are less reliable. It's less reliable because linear interpolations are likely to have error propagations for early years when gaps were filled using values in subsequent years, rather than interpolations using values in both previous and subsequent years. The validation should include more figures, for example, with y-axis as regression residuals and x-axis as years (when raw were available) for different crops.*

**Response**: Thanks for pointing this out! To address this comment, we added two time-series plots (Figure 2 and S3) to the supplementary. These plots show the residual, the inventory-based cropland area minus the rebuilt-map-based cropland area (Equation 7), and the relative residual, the ratio of residual to the inventory crop area (Equation 8), for each crop across all counties throughout the entire period. Examining Figure 2, it is evident that the errors are generally smaller than 0.2 Kha for the majority counties (>75%) across all years for nine crop types. Relatively greater residuals are observed in spring wheat, durum wheat, and rice before 1875 (Figure 2d, g, and i), which might be attributed to the marginal area of these three crops during the early years. Similarly, the relative residuals in most counties remain within ±2% for different crops, except for spring wheat, durum wheat, and rice before 1875 (Figure S3d, g, and i). Supplementing these findings with the 1:1 line comparison, our results demonstrate a robust alignment between the crop-specific plant area derived from maps and the inventory data. Detailed information was added to lines 241-250 and 272-278. For the period of 1850-1984, HYDE (available every 10 years) was initially used to identify the cropland distribution by calculating the fraction of cropland to the physical area for each grid. We then linearly interpolated the fraction for the missing years between two available years (every 10 years). The aims of the linear interpolation of HYDE are 1) identifying the cropland grids (fraction > 0) and 2) capturing the spatial variability of cropland fractions. The fractions of all grids within each county were further adjusted to match the inventory data. Therefore, the linear

interpolation approach used on HYDE has marginal impacts on the plant acreage but might influence the cropland distribution. We have revised our manuscript to provide a clear explanation of this process in lines 133-136 and the Supplementary methods: (2) Linear interpolation in HYDE.

***Lines 241-250***: Here, we adopted multiple indexes to evaluate the crop area discrepancy between the reconstructed maps and inventory data at various scales. At the county level, we utilized the residual ($resd_{ij}$) and relative residual ($relative\_resd_{ij}$) to describe the crop area difference and relative difference between the rebuilt maps and the inventory data (Equation 6 and 7). In addition, at the national scale, the Root Mean Squared Error ($RMSE$) and R-squared ($R^2$) are used to assess the crop area consistency between the crop maps and the inventory data.

$$resd_{ij} = inv_{ij} - map_{ij}, \tag{6}$$

$$relative\_resd_{ij} = (inv_{ij} - map_{ij}) * 100/inv_{ij}, \tag{7}$$

Where, $inv_{ij}$ and $map_{ij}$ are the crop area derived from the inventory data and the rebuilt maps at year $i$ and in county $j$, respectively. $resd_{ij}$ and $relative\_resd_{ij}$ are the residue and relative residue at year $i$ and in county $j$, respectively.

***Lines 272-278***: We further examined the time-series residual between the inventory data and maps (Figure 2 and S3). It is evident that the residuals (the inventory-based crop area minus the rebuilt-map-based crop area (Equation 7)) are generally smaller than 0.2 Kha for the majority counties (>75%) across all years for nine crop types. Relatively greater residuals are observed in spring wheat, durum wheat, and rice before 1875 (Figure 2d, g, and i), which might be attributed to the marginal area of these three crops during the early years. Similarly, the relative errors (the ratio of residual to the inventory crop area (Equation 8)) in most counties remain within ±2% for different crops, except for spring wheat, durum wheat, and rice before 1875 (Figure S3d, g, and i).

***Lines 133-136***: Consequently, HYDE (available every 10 years) was initially used to identify the cropland distribution by calculating the fraction of cropland to the physical area for each grid. We further linearly interpolated the fraction for the missing years between two available years to provide a potentially continuous cropland distribution (more details are presented in (2) Linear interpolation in HYDE of Supplementary Methods).

***Supplementary Methods:***

**(2) Linear interpolation in HYDE**

Here, the linear algorithm (Equation S2) is used to interpolate the potential cropland map in years when HYDE was unavailable before 1985.

$$A_{x,y}^{t_i} = \frac{A_{x,y}^{t_2} - A_{x,y}^{t_1}}{t_2 - t_1} \times (t_i - t_1) + A_{x,y}^{t_1} \tag{S2}$$

Where,

$t_2$ and $t_1$ are two adjacent years when HYDE is available, assuming $t_2$ is greater than $t_1$;

$t_i$ is any year between $t_1$ and $t_2$;

$A_{x,y}^{t_2}$, $A_{x,y}^{t_1}$ are the HDYE cropland percentage for the location (x, y) in year $t_2$ and $t_1$, respectively.

$A_{x,y}^{t_i}$ is the interpolated cropland percentage in the year $t_i$ at (x, y).

[Figure]

Figure 1. The distribution of residual (the inventory-based crop area minus the rebuilt-map-based crop area, defined by Equation 6) between the rebuilt inventory and maps from 1850 to 2021 (Kha is a thousand hectares). In each year, "Min-Max", "Median", and "25%-75%" reflects the extent of residual from all counties at levels of "minimum value to maximum value", "50th percentile", and "25th percentile to 75th percentile", respectively, which are corresponding to five percentiles in a box plot.

[Figure]

Figure S3. The distribution of relative residual (the ratio of the residual to the inventory crop area, defined by Equation 7) between the rebuilt inventory and rebuilt maps from 1850 to 2021. In each year, "Min-Max", "Median", and "25%-75%" reflects the extent of residual at levels of "minimum value to maximum value", "50th percentile", and "25th percentile to 75th percentile", respectively, which are corresponding to five percentiles in a box plot. The counties with cropland areas less than 1kha are excluded to avoid the case with a relative residual greater than 100%.

*2. In the linear-regression based validation, should clarify the dependency for derived vs. compared values. If the derived values are highly dependent on the compared values, no wonder high correlations were found.*

**Response**: We thank the reviewer for pointing this out. We utilized the planting area of different crops from inventory data as a benchmark to refine the gridded map. Consequently, in an ideal scenario, the cropland area derived from maps should align precisely with the inventory data. However, achieving exact alignment for every crop within each county is challenging, primarily due to constraints related to the limited cropland for allocating to different crop types. We employed Figure S2 (the original Figure 2) to assess the consistency between cropland areas from these two data across crops. We improved the explanation for the linear-regression based validation in lines 263-266 and the title of Figure S2.

***Lines 263-266:*** In this study, we adopted the inventory data to refine the gridded map, recognizing that achieving exact alignment for each crop type within each county might be challenging due to constraints related to the limited cropland area available for allocation. Here, we examined the crop-specific area alignment between the inventory data and our map products at multiple scales.

***The title of Figure S2 (the original Figure 2):*** Figure S2. Comparison of crop-specific cropland area between reconstructed maps and raw inventory data at county level in 1920, 1960, 2000, and 2020 (Kha is a thousand hectares). The point in subfigures represents the paired cropland area from the reconstructed map and raw inventory data for a certain county and year. The color bar in each subfigure indicates the probability density of paired point calculated by the gaussian kernel.

*3. The method uses 7 different datasets (Table 1). These datasets have duplicated items, like county-level crop areas that are existing/derivable in most datasets. Should provide information about the discrepancies of these duplicated items in different datasets, and how these discrepancies were considered in the used method. For example, when there are multiple choices, should justify why those particular datasets were selected as the basis for the linear interpolations. This is related to Minor Comments 9, 11, 15, 16, 17.*

**Response**: We appreciate and agree with the reviewer's constructive comments. Our primary objective is to reconstruct long-term gridded crop-specific cropland use maps. The long-term inventory of county-level crop-specific planting area serves as crucial information to adjust the spatial maps in terms of acreage. Three datasets we used in this study play different roles in reconstructing this inventory data. In our methodology, USDA-NASS Quickstat is the only dataset that reports crop-specific planting area at the county level. However, it comes with certain limitations, such as covering relatively shorter periods (e.g., 1953-2021 for rice) (Table S2). We used county-level crop-specific harvest area, provided by USDA-NASS Quickstat as well, to extend the period to the earlier date and gap-filled the missing years. In instances where the county-level harvested area data is unavailable, we relied on the state-level crop-specific planting area and harvested data instead. Finally, in this study, we explicitly reconstructed the plant area of 9 major crops and calculated the plant area for all other crops by obtaining the difference between the total plant area and the summation of 9 major crops.

To enhance clarity and consistency throughout the manuscript, we have revised terminology, unifying terms such as cropland area, crop area, crop-specific area, plant area, and harvest area to plant area and harvest area. We also added a more detailed description for the interpolation approach. The revision can be found in lines 145-182: Section 2.2 Reconstructing historical crop acreage at the county level.

***Lines 145-182***: 2.2 Reconstructing crop acreage history at the county level

[revised manuscript text omitted]

*4. In Figs. 8 and 9, the derived maps have apparent visual differences with HYDE. Explanations are needed for the differences, and for the impacts of the differences given that HYDE was used as input to derive the maps (Fig. 1). Can their regressions be quantified in validations?*

**Response**: We thank for the reviewer's suggestion. HYDE primarily served as a tool for identifying cropland grids and assessing the intensity of cropland per grid. This information was further adjusted based on the county-level inventory data from the US, leading to visual differences between our map and HYDE. For example, the inventory data shows that there was no cropland distributed in certain counties in North Dakota in 1850 (Figure 8). Consequently, we adjusted the grids with cropland in HYDE to zero for these specific counties. We have added explanations and the corresponding quantitative analysis in lines 419 to 428.

**Lines 419-428**: Spatially, we observed that HYDE exhibits broader cropland extent and a higher fraction of cropland per grid than our products, particularly in regions with low-density cropland distribution, such as the Northwest, Southeast, and Southwest (Figure 8 and Figure 9). This disparity might be attributed to the definition of cropland in HYDE, which includes both arable land and permeant cropland (Goldewijk, 2001) while our map exclusively accounts for crop planting area of crops. More importantly, the crop planting area of our map was constrained based on county level inventory data. Meanwhile, HDYE spatialized the subnational level inventory data to allocate cropland area to each grid in accordance with "cropland suitability maps" informed by dynamical social (historical population density) and stable environmental (soil suitability, temperature, and topography) information (Klein Goldewijk et al., 2011; Yu and Lu, 2018). As a result, greater acreage and wider extent of cropland were estimated by HYDE and were allocated to each grid (Figure 7, Figure S8, and Figure S9).

Minor Comments:

*1. May consider changing "land use and cover changes" to "land cover land use change (LCLUC)", which is a more used standard term in my experience.*

**Response**: We appreciate the reviewer's suggestion. We have changed this term to "land cover/land use change (LCLUC)" in lines 7-8.

**Lines 7-8**: Agricultural activities have been recognized as an important driver of land cover/land use change (LCLUC) and have significantly impacted the ecosystem feedback to climate by altering land surface properties.

*2. L50. "History database of global environment (HYDE) (Goldewijk et al., 2017) dataset provides the cropland area in each grid cell from 1000 BC to 2017 AD at a resolution of 5 arc-min. Similarly, Zumkehr and Cambell (2013) developed a cropland distribution dataset at a 5 arc-min resolution from 1850 to 2000."*

*- Should provide more information that how the early maps were made. That's the information many readers want to know.*

*-Should also clarify with references that the crop areas obtained with different approaches can be meaningfully combined to make historical maps, such as USDA Quickstat compared with satellite-derived NLCD, LCMAP, and CDL. Some datasets may have consistent over-estimates or under-estimates.*

**Response**: We are grateful for the valuable comments. We have added more details from the perspective of data and method to describe how the early maps in HYDE and ZCMAP were made in lines 52-57. In addition, we clarified the discrepancy among different sources of datasets and the rationality of merging inventory information into gridded maps by conducting a series of pre-processes for data fusion (more details in lines 116-136).

*Lines 52-57*: For instance, History database of global environment (HYDE) (Goldewijk et al., 2017) constructed a weighting algorithm involving dynamical social (historical population density and national/sub-national crop statistics, state level crop inventory in US) and stable environmental (soil suitability, temperature, and topography) factors to reconstruct the historical crop distribution at the resolution of 5 arc-minute. Similarly, Zumkehr and Cambell (2013) adopted a land-use model of Romankutty and Foley (Ramankutty and Foley, 1999) and a satellite-derived cropland distribution map to calculate the historical crop area grid by grid under the control of crop inventory records.

*Lines 116-136*: We compared the planting area between inventory data and CDL for nine crop types across counties from 2010 to 2021 (Figure S1). For most counties (>75%), the residuals (the inventory-based crop area minus CDL-based crop area) are less than 10 Kha for durum wheat while they are less than 5 Kha for other crops. NLCD and LCMAP, both derived from Landsat images with a resolution of 30m×30m, were integrated to provide the spatial information of cropland distribution from 1985 to 2009. NLCD crop area is highly consistent with CPAS and COA, except that the crop area was significantly underestimated in NLCD 1992 (Figure 4 in Yu and Lu, 2018), so it was excluded for reconstructing historical crop maps (Johnson, 2013). Due to its consistency in cropland area, we utilized NLCD for identifying the spatial distribution of cropland (Homer et al., 2020). However, NLCD provides around 5-year cyclical land cover maps from 2001 to 2019 (Homer et al., 2020). LCMAP offers annual land use data from 1985 to 2021. LCMAP adopts Anderson Level I-based legend, grouping cropland and pasture into one category (Xian et al., 2022). In contrast, NLCD uses a Level II-based legend where cropland and pasture are separately tracked (Xian et al., 2022) (Table S4). To generate a reliable cropland distribution, the long-term non-crop trajectory derived from NLCD was used to exclude all grids identified as cropland the LCMAP map (more details are presented in Supplementary Methods: (1) Preprocesses for LCMAP). For the period of 1850-1984, although both ZCMAP and HYDE provide the cropland distribution, HYDE considers the impacts of various environmental factors (soil suitability, temperature, and topography) on crop distribution compared with ZCMAP (Goldewijk, 2001; Goldewijk et al., 2011; Goldewijk et al., 2017; Zumkehr and Campbell, 2013). Consequently, HYDE (available every 10 years) was initially used to identify the cropland distribution by calculating the fraction of cropland to the physical area for each grid. We further linearly interpolated the fraction for the missing years between two available years to provide a potentially continuous cropland distribution (more details are presented in (2) Linear interpolation in HYDE of Supplementary Methods).

*3. L53. "In contrast, the resolution of Cropland Data Layer (CDL), National Land Cover Database (NLCD), and Land Change Monitoring, Assessment, and Projection (LCMAP) is down to 30m."*
*-Need refs.*

**Response**: Thanks for this suggestion. We have added references in lines 60-63.

***Line 60-63***: For instance, Cropland Data Layer (CDL), National Land Cover Database (NLCD), and Land Change Monitoring, Assessment, and Projection (LCMAP) provide the gridded cropland distribution maps at the resolution of 30m by 30m (Homer et al., 2020; Xian et al., 2022; Lark et al., 2017).

*4. L55. "However, their availability and continuity (available in the recent 40 years) are unable to provide historical cropland change patterns."—To here*

*-The statement regarding the 40-year availability of crop maps is ambiguous. It seems to be only about LCMAP, but the previous text also talks about NLCD and CDL.*

**Response**: We appreciate this suggestion. We have modified this sentence in lines 63-64.

***Lines 63-64***: However, these high-resolution datasets lack the capability to depict historical cropland change patterns before the emergence of satellite images.

*5. L56. "The more recent studies, such as Cao et al. (2021) and Li et al. (2023), developed long-term LUCC datasets at 1 km by 1 km resolution… Monfreda et al. (2008) and Tang et al. (2023) generated a global crop type map with more than 170 crop types in the year of 2000 and 2020, and CDL provides the annual crop type distribution in the conterminous US with more than 50 crop types from 2008 to now".*

*-Should provide information on how these datasets were made.*

**Response**: We are grateful for this suggestion. We have added more information about how these datasets were made in lines 64-72.

***Lines 64-72***: Recently, Cao et al. (2021) harmonized cropland demands from HYDE and Land-Use Harmonization 2 datasets with the combination of cropland suitability, kernel density, and other constraints to generate a cropland dataset from 10000 BCE to 2100 CE. Li et al. (2023) integrated an artificial neural network-based probability of occurrence estimation tool and multiple inventories to generate the historical cropland maps at the resolution of 1km by 1km. However, the crop type details are still missing in these datasets, making it challenging to identify the specific crop type change over space and time. On the other hand, Monfreda et al. (2008) combined a global cropland dataset and multi-level census statistics (national, state, and county) to generate a map depicting the area and yield of 175 crops circa the year 2000 around the world, and Tang et al. (2023) further updated it to depict 173 crops circa the year 2020.

*6. Fig. 1. What's the purpose of deriving state-level information, given that county-level information is derived?*

**Response**: Here, the cropland area trend from the generated state-level information works as a referenced trend to gap-fill the total county planting area from 1850 to 2021. Please see the lines 169-171.

***Lines 169-171***: We gap-filled the total county planting area from 1850 to 2021 by using state total planting area as a referenced trend (using Equation 1 for gap-filling in cases where only beginning or ending year is available and Equation 2 in cases where both beginning and ending years are known).

*7. Table 1. "Linearly interpolation" -> "Linear interpolation".*

**Response**: Thanks for this suggestion. We have changed it to "Linear interpolation" in Table 1.

*8. Table 1. What are the differences between linear interpolation and gap-filling in missing years?*

**Response**: We adopted a variety of interpolation approaches according to the availability of data. For the raw data that does not have any correlated references, we used linear interpolation. For example, HYDE offers spatial maps during 1850-1985 at 10-year interval. Due to the lack of other references, we applied linear interpolation to fill the missing years within each 10-year period (More details presented in Supplementary method: (2) Linear interpolation in HYDE). Otherwise, we applied different gap-filling approaches correspondingly. We specified the approaches by referring to equations in our manuscript to avoid confusion (Equation 1 or 2). One example is that we used the crop-specific harvested area as a reference trend to gap-fill the crop-specific planting area for each county using Equation 2. Details are lines 154-156, 162-166, 169-171, and 173-175.

***Lines 154-156***: We used the interannual variations of arable land of each state extracted from HYDE to extrapolate the total planting area from 1908 to 1850 (Equation 1).

***Lines 162-166***: For the period that the harvesting areas are unavailable, we interpolated the planting area from 1850 to 2021 based on the total planting area generated above as a referenced trend. Equation 1 was used when only the beginning or the ending year of the period is available, while Equation 2 was used when both beginning and ending years are available. The planting area of "others" was obtained by calculating the difference between the total planting area and the summation of planting area of 9 major crops.

***Lines 169-171***: We gap-filled the total county planting area from 1850 to 2021 by using state total planting area as a referenced trend (using Equation 1 for gap-filling in cases where only beginning or ending year is available and Equation 2 in cases where both beginning and ending years are known).

***Lines 173-175***: For the period when harvesting areas are unavailable, we gap-filled the planting areas of each crop during 1850-2021 based on the state-level crop-specific planting area generated above as a referenced trend (Equation 1 and 2).

***Supplementary method:***

**(2) Linear interpolation in HYDE**

Here, the linear algorithm (Equation S2) is used to interpolate the potential cropland map in years when HYDE was unavailable before 1985.

$$A_{x,y}^{t_i} = \frac{A_{x,y}^{t_2} - A_{x,y}^{t_1}}{t_2 - t_1} \times (t_i - t_1) + A_{x,y}^{t_1} \tag{S2}$$

Where,

$t_2$ and $t_1$ are two adjacent years when HYDE is available, assuming $t_2$ is greater than $t_1$;

$t_i$ is any year between $t_1$ and $t_2$;

$A_{x,y}^{t_2}$, $A_{x,y}^{t_1}$ are the HDYE cropland percentage for the location (x, y) in year $t_2$ and $t_1$, respectively.

$A_{x,y}^{t_i}$ is the interpolated cropland percentage in the year $t_i$ at (x, y).

*9. Table 1. NASS-CPAS, NASS-COA, and USDA-NASS have duplicated information, like state-level harvest areas? Do the duplicated items have same values across the three datasets?*

**Response**: We thank the reviewer for pointing out the unclear description of these three datasets. These three datasets provide different information regarding cropland areas. NASS-CPAS reports the annual state-level total planting area of most principal crops but excludes some minor crops. We obtained the total planting area of all crop types for each state from NASS-COA. However, it is only available every 4-5 years. We harmonized these two datasets to obtain the annual total planting area of all crops for each state. On the other hand, both NASS-COA and USDA-NASS Quickstat provides planting and harvesting areas at the state- and county level. NASS-COA only offers the total area of all crops, while USDA-NASS Quickstat offers the crop-specific area. We added more details to explain their differences in lines 148-151, 156-157, 168-169, and 171-173.

*Lines 148-151*: NASS-CPAS reports the annual total planting area of major crops for each state from 1909 to 2021. However, some minor crop types, such as vegetables and fruits, are excluded. USDA-COA provides the total areas of crop harvest, failure, and fallow for each state from 1925 to 2017 with 4~5-year intervals.

*Lines 156-157*: To identify the planting acreage change for nine major crop types, we obtained the state-level crop-specific harvesting and planting area from USDA-NASS Quickstat.

*Lines 168-169*: USDA-COA reports the total county cropland area from 1925 to 2017 with 4~5-year intervals.

*Lines 171-173*: Similar to the state-level crop-specific planting area, we converted the harvesting areas to planting areas of nine major crops in each county from USDA-NASS Quickstat, with varied availability (Table S1).

*10. L116. "NASS-CPAS reports the annual plant area of all principal crops for each state from 1909 to 2021"*

*-This is inconsistent with the information in Table 1 that NASS-CPAS provides "State-level total planted area".*

**Response**: We appreciate the reviewer for pointing out this unclear description. We have revised "State-level total planting area" to "State-level total planting area of major principal crops" in Table 1.

*11. L119. "We computed the difference between these two datasets for available years and linearly interpolated unavailable years during 1909-2021. The interpolated difference was added to NASS-CPAS to generate the annual state-level total crop plant area from 1909 to 2021. We used the interannual variations of arable land of each state extracted from HYDE to interpolate the total planting area during 1850-1908 (Equation 1)."*

*-"difference between these two datasets" regarding state-level crop area?*

*-How big are the differences, e.g. regarding relative percentage between NASS-CPAS and HYDE? See Major Comment 3.*

*-Why not use USDA-NASS Quickstat? See Major Comment 3.*

**Response**: The difference between these two datasets refers to the total plant area of all minor crops. NASS-CPAS offers the annual total planting area of all major crops, which excludes some minor crops such as vegetables and fruits. NASS-COA provides the total planting area (sum of harvest, failure, and fallow) of all crops but at 4~5-year intervals. To obtain the annual total planting area of all crops, we calculated the difference between these two datasets and linearly interpolated the unavailable years. The differences were then added back to NASS-CPAS. The total planting

area generated from CPAS and COP is further used as a reference to interpolate USDA-NASS Quickstat in harvesting area unavailable years. Please see details in lines 148-165. We assessed the differences between our map and HYDE. Please see our response in Major comment 4.

*Lines 148-165*: NASS-CPAS reports the annual total planting area of major crops for each state from 1909 to 2021. However, some minor crop types, such as vegetables and fruits, are excluded. USDA-COA provides the total areas of crop harvest, failure, and fallow for each state from 1925 to 2017 with 4~5-year intervals. We computed the difference between these two datasets for available years and linearly interpolated unavailable years during 1909-2021. The difference was assumed to be the planting area of those minor crops. The interpolated difference was then added back to NASS-CPAS to generate the annual state-level total crop planting area of all crops from 1909 to 2021. We used the interannual variations of arable land of each state extracted from HYDE to extrapolate the total planting area from 1908 to 1850 (Equation 1). To identify the planting acreage change for nine major crop types, we obtained the state-level crop-specific harvesting and planting area from USDA-NASS Quickstat. The available harvesting and planting areas vary among crop types and states, for which the harvesting areas usually have earlier-year reports than those of planting areas (Table S2). The harvesting area is highly correlated to planting area in terms of interannual variation. We calculated the ratio of planting area to harvesting area for the earliest available year of planting area. We then converted the harvesting areas to planting areas by timing the ratio with the harvesting areas to extend the planting areas to an earlier period. For the period that the harvesting areas are unavailable, we interpolated the planting area from 1850 to 2021 based on the total planting area generated above as a referenced trend. Equation 1 was used when only the beginning or the ending year of the period is available, while Equation 2 was used when both beginning and ending years are available.

*12. L128. "For the period that the harvest areas are unavailable, we interpolated the plant area from 1850 to 2021 based on the total cropland area generated above (Equation 1 and 2)".*
*-Unclear. Also Equations 1 and 2 are not clearly described.*
**Response**: We thank the reviewer for pointing out this unclear description. We have recognized this in lines 162-165.
*Lines 162-165*: For the period that the harvesting areas are unavailable, we interpolated the planting area from 1850 to 2021 based on the total planting area generated above as a referenced trend. Equation 1 was used when only the beginning or the ending year of the period is available, while Equation 2 was used when both beginning and ending years are available.

*13. L132. "We gap-filled the total county cropland from 1850 to 2021 by state total cropland area (Equation 1 and 2)".*
*-Unclear.*
**Response**: We appreciate the reviewer for point this unclear description. We have recognized this in lines 169-171.
*Lines 169-171*: We gap-filled the total county planting area from 1850 to 2021 by using state total planting area as a referenced trend (using Equation 1 for gap-filling in cases where only beginning or ending year is available and Equation 2 in cases where both beginning and ending years are known).

*14. L133. "Similar to the state-level crop-specific area, we converted the harvest areas to plant areas of 9 major crops in each county from USDA-NASS Quickstat, with varied availability (Table S1). For the period when harvest areas are unavailable, we gap-filled the plant areas during 1850-2021 based on the state-level crop-specific plant area generated above (Equation 1 and 2)."*

*-Clarify where "state-level crop-specific area" was obtained.*

*- Clarify "generated above" where?*

**Response**: We thank the reviewer for these suggestions. We described how to obtain the state-level crop-specific planting area in lines 147-166.

***Lines 147-166***: Our reconstruction process was initiated with the development of crop-specific planting areas at the state level. NASS-CPAS reports the annual total planting area of major crops for each state from 1909 to 2021. However, some minor crop types, such as vegetables and fruits, are excluded. USDA-COA provides the total areas of crop harvest, failure, and fallow for each state from 1925 to 2017 with 4~5-year intervals. We computed the difference between these two datasets for available years and linearly interpolated unavailable years during 1909-2021. The difference was assumed to be the planting area of those minor crops. The interpolated difference was then added back to NASS-CPAS to generate the annual state-level total crop planting area of all crops from 1909 to 2021. We used the interannual variations of arable land of each state extracted from HYDE to extrapolate the total planting area from 1908 to 1850 (Equation 1). To identify the planting acreage change for nine major crop types, we obtained the state-level crop-specific harvesting and planting area from USDA-NASS Quickstat. The available harvesting and planting areas vary among crop types and states, for which the harvesting areas usually have earlier-year reports than those of planting areas (Table S2). The harvesting area is highly correlated to planting area in terms of interannual variation. We calculated the ratio of planting area to harvesting area for the earliest available year of planting area. We then converted the harvesting areas to planting areas by timing the ratio with the harvesting areas to extend the planting areas to an earlier period. For the period that the harvesting areas are unavailable, we interpolated the planting area from 1850 to 2021 based on the total planting area generated above as a referenced trend. Equation 1 was used when only the beginning or the ending year of the period is available, while Equation 2 was used when both beginning and ending years are available. The planting area of "others" was obtained by calculating the difference between the total planting area and the summation of planting area of 9 major crops.

*15. What are the differences of crop areas between Quickstat and CDL for overlapping years?*

**Response**: Here, we added a plot (Figure S1) to show the discrepancy between USDA Quickstat and CDL among specific crop types from 2010 to 2021. The residuals (the inventory-based crop area minus the CDL-based crop area) between the rebuilt Quickstat and CDL are less than 5 Kha for most crop types, except for durum wheat, across most counties (>75%). After the further adjustment in the study, this relatively large error is decreased to <0.2 Kha for nine crop types (Figure 2). Please see lines 116-119.

***Lines 116-119***: We compared the planting area between inventory data and CDL for nine crop types across counties from 2010 to 2021 (Figure S1). For most counties (>75%), the residuals (the inventory-based crop area minus CDL-based crop area) are less than 10 Kha for durum wheat while they are less than 5 Kha for other crops.

[Figure]

Figure S1. The distribution of residual (the inventory-based crop area minus the CDL-based crop area, defined by Equation 6) between the USDA Quickstat and CDL from 2010 to 2021 (Kha is a thousand hectares). In each year, "10%-90%", "Median", and "25%-75%", reflect the extent of residual at levels of "10 percentile to 90th percentile", "50th percentile", and "25th percentile to 75th percentile", respectively.

*16. Fig. 1. For crop density maps, CDL was used for 2010-2021, and LCMAP + NLCD were used for 1985-2009. Why not use LCMAP + NLCD for the whole period 1985-2021?*

**Response**: The principal aim of the study is to retrieve historical crop type maps. Among these satellite-derived LCLUC datasets, CDL uniquely provides detailed crop type information, with CDL 2010 serving as the foundational map for historical crop type delineation. Other products such as LCMAP and NLCD provide cropland information without specifying crop type information. Consequently, CDL is prioritized for adoption when available, leading to its utilization for the period of 2010-2021.

*17. L147. "CDL, LCMAP, and HYDE were used to provide the potential cropland distribution in P2010, P1985, and P1850, respectively".*

*-Why not used LCMAP for P2010?*

**Response**: One major aim of this study is to reconstruct the historical crop type map. Only CDL provides the crop-specific information, while LCMAP only offers the total cropland distribution. Thus, CDL is prioritized for adoption when available, leading to its utilization for the period of 2010-2021.

*18. L164. "Taking developing the density map in the year 2009 as an example, we first calculated the annual difference in each grid from 2009 to 2010 based on the LCMAP density maps. Then, we applied that difference to the adjusted CDL 2010 map to generate the density map 2009 with keeping the cropland area consistent with the inventory area. Following the same rule, the adjusted LCMAP 1985 was used to retrieve the density maps in P1850."*

*-This text is incomprehensible.*

**Response**: We appreciate the reviewer for pointing out this unclear description. We have reorganized this description in lines 206-210.

***Lines 206-210***: Taking the year 2009 as an example, the interannual difference in each grid between LCMAP 2009 and 2010 was applied to the adjusted CDL 2010 to generate the potential crop density map in year 2009. Then, the potential density map was further corrected by the inventory data through Equation 3. Following the same rule, the difference between the interpolated HYDE 1985 and 1984 was applied to the adjusted LCMAP 1985 to retrieve the density maps in P1850.

*19. L175. "By integrating resampled crop type maps and reconstructed cropland density maps, we counted the total area for each type at the county level and identified specific crop types with a greater area than the inventory data. We further converted the surplus area from these types to other types (Equation 4 and 5). In particular, considering the natural planting scenario, the surplus area was randomly selected for converting to other types to avoid a grid planted by a fixed type."*

*-Incomprehensible, e.g. "identified specific crop types with a greater area than the inventory data".*

**Response**: We are grateful for pointing out this unclear description. We have reorganized this in lines 216-220.

***Lines 216-220***: By integrating resampled crop type maps and reconstructed cropland density maps, we counted the total area for each type at the county level, and identified the crop types whose area is greater than the corresponding inventory record. We further converted the surplus pixels from these types to other types whose area is less than inventory data (Equation 4 and 5). In particular, to avoid a grid planted by a fixed type for a long time, the surplus pixels are randomly selected for the conversion across different crop types.

*20. Fig. 4. The density maps may be related to the CONUS field size map in the following ref.*

Yan, L. and Roy, D.P., 2016. Conterminous United States crop field size quantification from multi-temporal Landsat data. Remote Sensing of Environment, 172, pp.67-86

**Response**: Yan and Roy (2016) pointed out that the high crop density area tends to be the large crop field size. We have added this reference in lines 308-310.

*Lines 308-310*: Generally, the hotspots of cropland are concentrated in the Midwest and Great Plains regions (the spatial pattern of US subregions showed in Figure 5(2-a)), starting from 1950, where large crop field sizes were likely to occur (Yan and Roy, 2016).

*21. In Fig. 7, should clarify what datasets were not used in the study? Also what's the purpose to show the datasets not used?*

**Response**: We appreciate this valuable comment. We have clarified the unused datasets in lines 465-466 and specified the objective of showing these datasets in lines 403-404.

*Lines 465-466*: In particular, YLMAP, ZCMAP, CONUS, and GBC are not used in this study.

*Lines 403-404*: We systematically compared our product with previous datasets regarding the historical total cropland area (Figure 7) and their spatial patterns (Figure 8) to provide a complete reference for potential applications.

---

## Author Comment (AC2)

**Reviewer#3**

Summary

Mapping the spatial distribution and temporal dynamics of crop area and crop types is essential for crop management. However, long-term and relatively high-spatial resolution crop area data with detailed crop type information is lacking. Here Ye et al., combined multi-sources of crop area and type information and generated the 1-km maps of crop area and types in the conterminous US during 1850-2021. The manuscript is generally well organized, and easy to follow. I only have few specific concerns listed below.

**Response:** We appreciate the reviewer's positive feedback and have revised our manuscript. Please see our response below.

Specific comments

*1)  Line 8, 'climate, air', there could have some overlaps between these two terms.*

**Response:** We appreciate this suggestion. We have changed "climate, air" to climate in lines 7-8.

*Lines 7-8*: Agricultural activities have been recognized as an important driver of land cover/land use change (LCLUC) and have significantly impacted the ecosystem feedback to climate by altering land surface properties.

*2)  Line 11, each time when I mention 'high resolution', I am always very cautious. In remote sensing, high resolution could be meter to submeter. So 'relatively high resolution' could be more rigorous.*

**Response**: We thank for this suggestion. We have revised "high resolution" to "relatively high resolution" line 11.

*Lines 11*: there remains a dearth of a relatively high-resolution dataset with crop type details over a long period

*3)  Line 85, any justification about the linear assumption?*

**Response**: We appreciate this valuable suggestion. After conducting a comprehensive data search, we discovered that the crop rotation information was recorded by the "Tailored Reports: Crop Production Practices" (USDA- Agricultural Resource Management Survey). We further extracted the state-level crop rotation information from 1996 to 2010, and found the rotation rate in each state kept relatively stable during ARMS-available years. Consequently, we adopted a new assumption that the rotation percentage between corn and soybean remained stable in each state when the crop rotation information was unavailable from 1940 to 2009. Furthermore, we updated the crop type maps under the new corn-soybean rotation assumption. More details about the rotation process are in lines 96-97, 230-239 and Table S3.

*Lines 96-97*: The rotation percentage between corn and soybean remained constant when the rotation information was unavailable from 1940 to 2009.

*Lines 230-239*: Considering the dominant crop rotation type in US, soybean and corn rotation, we simulated the corn-soybean rotation by randomly switching a certain area between corn and soybean according to the rotation rate. The crop rotation information from 1996 to 2010 at state level was documented by the "Tailored Reports: Crop Production Practices" of USDA's Agricultural Resource Management Survey (ARMS) (https://data.ers.usda.gov/reports.aspx?ID=17883). The rotation rate was calculated as the ratio of the sum of corn-soybean and soybean-corn acreage to the total area of corn and soybean. We found that the rotation rate in each state kept relatively stable in the ARMS-available years, and assumed that the rotation rate in the missing years is the same as the mean rate from available years (Table S3), which is further applied to corresponding counties. Because soybean was rarely planted in the Corn Belt before 1940 (Yu et al., 2018), we only considered the corn-soybean rotation during the period 1940-2009 in 17 states (Table S3) (Padgitt et al., 1990).

Table S1. The mean corn-soybean ration ratio calculated from the available years.

| State | Rotation Rate (%) | Standard Deviation | State | Rotation Rate (%) | Standard Deviation |
|---|---|---|---|---|---|
| IL | 85 | 5 | MO | 58 | 8 |
| IN | 81 | 10 | NB | 59 | 12 |
| IA | 86 | 5 | NC | 30 | 10 |
| KS | 35 | 9 | OH | 64 | 12 |
| KY | 58 | 15 | PA | 27 | 10 |
| LA | 16 | 10 | SD | 69 | 7 |
| MI | 49 | 11 | TN | 35 | 15 |
| MN | 74 | 6 | WI | 43 | 13 |
| MS | 14 | 9 | | | |

4) *Line 87, references are needed following 'diversity index'*

**Response**: Thanks for this suggestion. We have added a reference.

L Jost: Entropy and diversity, Opinion, 2, 363–375, 2006.

5) *Line 109-110: please also briefly mention how you conducted the filtering operation in the main text.*

**Response**: We are grateful for this advice. We added a brief description in lines 125-129.

*Lines 125-129*: LCMAP adopts Anderson Level I-based legend, grouping cropland and pasture into one category (Xian et al., 2022). In contrast, NLCD uses a Level II-based legend where cropland and pasture are separately tracked (Xian et al., 2022) (Table S4). To generate a reliable cropland distribution, the long-term non-crop trajectory derived from NLCD was used to exclude all grids identified as cropland the LCMAP map (more details are presented in Supplementary Methods: (1) Preprocesses for LCMAP).

*6) Table 1, how did you resample 30 m data to 1km, how did you interpolate the 5arc-min data to 1km?*

**Response**: Thanks for this suggestion. We added the description of resampling map in lines 136-141

*Lines 136-141*: All gridded datasets were resampled to 1km. We employed a 1km*1km window to aggregate the total cropland area from the 30m*30m map and assigned the area to the corresponding 1km*1km grid. To resample the CDL crop type map from 30m to 1km, the crop type in each 1km by 1km pixel was assigned to the dominant crop type with the largest fraction of land area within the 1km*1km window. Conversely, the cropland percentage in each 5 arc-min grid is interpolated to 1km*1km grid cells with an assumption that cropland percentage is evenly distributed within the 5 arc-min by 5 arc-min window.

*7) Line 110: why not use the data from Zumkehr and Cambell (2013). I assume differences exist between these two long-term datasets.*

**Response**: In reconstructing historical cropland maps, HDYE adopted a weighing map involving social and multiple environmental factors (soil suitability, temperature, and topography) to spatialize cropland area (Goldewijk, 2001; Goldewijk et al., 2011; Goldewijk et al., 2017), while Zumkehr and Cambell (2013) adopted a land-use model of Romankutty and Foley (1999) to calculate historical cropland grid area based on a satellite-derived map. Compared with ZCMAP, HYDE considers the impacts of various environmental factors on crop distribution. Consequently, we adopted HYDE products as potential cropland distributions to retrieve historical cropland maps. Please see lines 129-134.

*Lines 129-134*: For the period of 1850-1984, although both ZCMAP and HYDE provide the cropland distribution, HYDE considers the impacts of various environmental factors (soil suitability, temperature, and topography) on crop distribution compared with ZCMAP (Goldewijk, 2001; Goldewijk et al., 2011; Goldewijk et al., 2017; Zumkehr and Campbell, 2013). Consequently, HYDE (available every 10 years) was initially used to identify the cropland distribution by calculating the fraction of cropland to the physical area for each grid.

*8) Line 119-121: please briefly explain what do the differences between NASS-CPAS and USDA-COA mean in the main text.*

**Response**: We appreciate this suggestion. We have explained the difference between NASS-CPAS and USDA-COA in lines 148-153.

*Lines 148-153*: NASS-CPAS reports the annual total planting area of major crops for each state from 1909 to 2021. However, some minor crop types, such as vegetables and fruits, are excluded. USDA-COA provides the total areas of crop harvest, failure, and fallow for each state from 1925 to 2017 with 4~5-year intervals. We computed the difference

between these two datasets for available years and linearly interpolated unavailable years during 1909-2021. The difference was assumed to be the planting area of those minor crops.

9)  Line 122: 'interpolate'->'extrapolate'

**Response**: We appreciate the reviewer's suggestion. We have revised it in lines 154-156.

*Lines 154-156*: We used the interannual variations of arable land of each state extracted from HYDE to extrapolate the total planting area from 1908 to 1850 (Equation 1).

10)  Fig. 2, Please explain what does each point mean in the caption.

**Response**: We thank for this suggestion. We moved Figure 2 to Figure S2, and added the explanation in the title of Figure S2.

*The title of Figure S2*: Comparison of crop-specific cropland area between reconstructed maps and raw inventory data at county level in 1920, 1960, 2000, and 2020 (Kha is a thousand hectares). The point in subfigures represents the paired cropland area from the reconstructed map and raw inventory data for a certain county and year. The color bar in each subfigure indicates the probability density of paired point calculated by the gaussian kernel

11)  Fig. 2, In addition to this kind of scatter plot, how about plotting the time series of crop-specific cropland area between reconstructed maps and raw inventory data to show that the reconstructed maps captured the interannual variations/trend of the raw data.

**Response**: We appreciate this valuable suggestion. To address this comment, we added two additional figures (Figure 2 and S3) to show the historical cropland residual and relative residual between maps and the inventory data at county level. The results show that the residual (the inventory-based crop area minus the rebuilt-map-based crop area) in planting area between the inventory data and maps is less than 0.2 Kha, and the relative residual (the ratio of residual to the inventory crop area) to the plant area is less than 10% for most counties (>75%) throughout the entire study period for nine major crop types, which indicate that the developed maps have strong capacity to capture the historical cropland area change. More details are presented in lines 243-248.

*Lines 243-248*: We further examined the time-series residual between the inventory data and maps (Figure 2 and S3). It is evident that the residuals (the inventory-based crop area minus the rebuilt-map-based crop area (Equation 7)) are generally smaller than 0.2 Kha for the majority counties (>75%) across all years for nine crop types. Relatively greater residuals are observed in spring wheat, durum wheat, and rice before 1875 (Figure 2d, g, and i), which might be attributed to the marginal area of these three crops during the early years. Similarly, the relative errors (the ratio of

residual to the inventory crop area (Equation 8)) in most counties remain within ±2% for different crops, except for spring wheat, durum wheat, and rice before 1875 (Figure S3d, g, and i).

[Figure]

Figure 1. The distribution of residual (the inventory-based crop area minus the rebuilt-map-based crop area, defined by Equation 6) between the rebuilt inventory and maps from 1850 to 2021 (Kha is a thousand hectares). In each year, "Min-Max", "Median", and "25%-75%" reflects the extent of residual from all counties at levels of "minimum value to maximum value", "50th percentile", and "25th percentile to 75th percentile", respectively, which are corresponding to five percentiles in a box plot.

[Figure]

Figure S1. The distribution of relative residual (the ratio of the residual to the inventory crop area, defined by Equation 7) between the rebuilt inventory and rebuilt maps from 1850 to 2021. In each year, "Min-Max", "Median", and "25%-75%" reflects the extent of residual at levels of "minimum value to maximum value", "50th percentile", and "25th percentile to 75th percentile", respectively, which are corresponding to five percentiles in a box plot. The counties with cropland areas less than 1kha are excluded to avoid the case with a relative residual greater than 100%.

**References:**

Goldewijk, K. K.: Estimating global land use change over the past 300 years: The HYDE database, Global Biogeochem. Cycles, 15, 417–433, https://doi.org/10.1029/1999GB001232, 2001.

Goldewijk, K. K., Beusen, A., Doelman, J., and Stehfest, E.: Anthropogenic land use estimates for the Holocene - HYDE 3.2, Earth Syst. Sci. Data, 9, 927–953, https://doi.org/10.5194/essd-9-927-2017, 2017.

Klein Goldewijk, K., Beusen, A., van Drecht, G., and de Vos, M.: The HYDE 3.1 spatially explicit database of human-induced global land-use change over the past 12,000 years, Glob. Ecol. Biogeogr., 20, 73–86, https://doi.org/https://doi.org/10.1111/j.1466-8238.2010.00587.x, 2011.

Ramankutty, N. and Foley, J. A.: Estimating historical changes in land cover North American croplands from 1850 to 1992, Glob. Ecol. Biogeogr., 8, 381–396, https://doi.org/10.1046/j.1365-2699.1999.00141.x, 1999.

Yu, Z., Lu, C., Cao, P., and Tian, H.: Long-term terrestrial carbon dynamics in the Midwestern United States during 1850-2015: Roles of land use and cover change and agricultural management, Glob. Chang. Biol., 24, 2673–2690, https://doi.org/10.1111/gcb.14074, 2018.

Zumkehr, A. and Campbell, J. E.: Historical U.S. cropland areas and the potential for bioenergy production on abandoned croplands, Environ. Sci. Technol., 47, 3840–3847, https://doi.org/10.1021/es3033132, 2013.